



# A Mechanistic Analysis of Tropical Pacific Dynamic Sea Level in GFDL-OM4 under OMIP-I and OMIP-II Forcings

Chia-Wei Hsu[1], Jianjun Yin[1], Stephen M. Griffies[2,3], and Raphael Dussin[2]

[1]University of Arizona, Department of Geoscience, 1040 E 4th St, Tucson, AZ 85721, USA
[2]NOAA Geophysical Fluid Dynamics Laboratory, Princeton USA
[3]Princeton University Atmospheric and Oceanic Sciences Program, Princeton USA

**Correspondence:** Chia-Wei Hsu (chiaweih@arizona.edu)

**Abstract.** The sea level over the tropical Pacific is a key indicator reflecting vertically integrated heat distribution over the ocean. Here we use the Geophysical Fluid Dynamics Laboratory OM4 (GFDL-OM4) global ocean-sea ice model forced by both the CORE and JRA55-do atmospheric states (OMIP-I and OMIP-II) to evaluate the model performance and biases compared against available observations. We find persisting mean state dynamic sea level (DSL) bias along 9°N even with updated wind forcing in JRA55-do relative to CORE. The mean state bias is related to biases in wind stress forcing and geostrophic currents in the 4°N to 9°N latitudinal band. The simulation forced by JRA55-do significantly reduces the bias in DSL trend over the northern tropical Pacific relative to CORE. In the CORE forcing, the anomalous westerly wind trend in the eastern tropical Pacific causes an underestimated DSL trend across the entire Pacific basin along 10°N. The simulation forced by JRA55-do significantly reduces the bias in DSL trend over the northern tropical Pacific relative to CORE. We also identify a bias in the easterly wind trend along 20°N in both JRA55-do and CORE, thus motivating future improvement. In JRA55-do, an accurate Rossby wave initiated in the eastern tropical Pacific at seasonal time scale corrects a biased seasonal variability of the northern equatorial counter-current in the CORE simulation. Both CORE and JRA55-do generate realistic DSL variation during El Niño. We find an asymmetry in the DSL pattern on two sides of the equator is strongly related to wind stress curl that follows the sea level pressure evolution during El Niño.

## 1 Introduction

A key goal for the Coupled Model Inter-comparison Project Phase 6 (CMIP6), including the CMIP6 Ocean Model Inter-comparison Project (OMIP), is to determine and to understand systematic model biases compared with observations, including both internal climate variability and externally-forced changes (Eyring et al., 2016; Griffies et al., 2016). In this paper, we focus on model biases found in OMIP simulations of the tropical Pacific, defined as the region within the 20°S-20°N zonal band inside the Pacific basin. This region is characterized by some of the most significant sea level variability on interannual time scales (up to several hundreds of millimeters). The nations in this region are highly affected by such variations as well as long-term trends, thus making a systematic analysis of tropical Pacific sea level biases important both societally and scientifically.

Sea level in the tropical Pacific reflects ocean heat content variability and long-term trends. Given the large size of this oceanic region, there is a high correlation between tropical Pacific sea level and global mean surface temperature at interannual





and longer time scales (Trenberth, 2002; Peyser et al., 2016; Hamlington et al., 2020). Hence, an accurate simulation of the tropical Pacific sea level variability and evolution supports a mechanistic understanding of climate variability and trends using forced ocean models, as well as their prediction using coupled climate models.

Studies of the mean state and seasonal cycle of tropical Pacific sea level started well before the routine availability of satellite altimetry measurements and realistic global climate models (Wyrtki, 1974). Some studies focused on the sea level

changes during one or two specific El Niño events due to the uniqueness and the available sea level observation from tide gauges (Cane, 1984; Busalacchi and Cane, 1985). In the current study, we use a CMIP6/OMIP ocean climate model with eddy-permitting grid spacing forced by OMIP-I and OMIP-II surface atmospheric fields. Comparisons to satellite altimeter measurements support an understanding of the processes underlying tropical Pacific sea level variability and trends. Recent studies have shown improved model performance due to updated surface forcing, with particular improvements in the wind

field (Taboada et al., 2018), and from refined grid spacing and improved model numerics and physics (Tsujino et al., 2020; Adcroft et al., 2019). Griffies et al. (2014) studied model performance and sea level biases for CMIP5-era ocean climate models forced by OMIP-I (CORE). The authors identified model limitations in simulating the sea level variation patterns by using the same atmospheric dataset with different ocean models. However, a detailed analysis of the mechanisms for simulation biases was lacking. Here, we take a complementary approach by performing a detailed analysis of one ocean model forced by OMIP-

I (CORE) and OMIP-II (JRA55-do), allowing for a detailed investigation of the causes for such model biases. We focus on tropical Pacific sea level variability and trends across different time scales, with an understanding of interannual variability in this region of particular importance for attributing observed sea level changes according to natural and anthropogenic effects.

In Section 2, we describe the model and experimental design for the simulations, list the observational data used to evaluate the simulations, and present the overall analysis framework. In Section 3, we present an analysis of the time-mean Pacific sea

level patterns and the associated biases. Through the heat budget analyses we find the mean state bias of heat advection and boundary fluxes cannot explain the sea level trend bias in simulations forced by OMIP-I and OMIP-II. An analysis is presented in Section 4 using decadal and longer trends to determine dominant factors causing the sea level trend bias. Besides the time-mean and long-term trend, sea level variability over seasonal and interannual time scale can also lead to significant change in the tropical Pacific. An assessment of seasonal sea level variability and interannual El Niño variability is presented in Section

5 and 6, respectively. The conclusion of our analysis and recommended key bias to correct in future simulations are offered in Section 7.

## 2   Model simulations, observational data, and analysis framework

Here we describe the general circulation model used in this study; the experimental design for the simulations; the observational-based datasets used to evaluate the simulations; and the methodology used to conceptually frame our analysis.



## 2.1 Ocean model and atmospheric forcing

We use the GFDL-OM4 ocean-sea ice model as documented by Adcroft et al. (2019), with OM4 having an eddy-permitting grid spacing of 0.25°. This relatively fine resolution grid is especially suitable for our study of tropical Pacific sea level (defined as the region within the 20°S-20°N zonal band inside the Pacific basin) given that it simulates boundary currents and ocean eddies better than the more commonly used one-degree class of models. Furthermore, OM4 makes use of a hybrid geopotential-isopycnal vertical coordinate (Chassignet et al., 2003; Adcroft et al., 2019), which is particularly useful in maintaining a realistic tropical Pacific thermocline whereas many $z$-coordinate models have an overly diffuse thermocline (Tseng et al., 2016; Griffies et al., 2009).

We force OM4 with the two atmospheric datasets used for the Ocean Model Inter-comparison Project (OMIP) versions I and II (Griffies et al., 2016; Tsujino et al., 2020). OMIP-I is forced by the CORE dataset of Large and Yeager (2009) and it extends over years 1948-2007 (hereafter, CORE), whereas OMIP-II uses the JRA55-do dataset of Tsujino et al. (2018), which extends over years 1958-2018. The OMIP protocol is detailed in Griffies et al. (2016) and Tsujino et al. (2020), with the use of two forcing datasets allowing us to assess robustness of simulated features and to better attribute biases. Following Tsujino et al. (2020), all simulations are spun-up by running for five cycles of the respective forcing datasets, over which time the upper ocean reaches a quasi-equilibrium state. We then present our analysis for the sixth forcing cycle.

## 2.2 Observationally based datasets

Total sea level changes can be derived from satellite altimetry. The Copernicus Marine Environment Monitoring Service (CMEMS) provides a gridded sea level dataset by combining multi-mission altimeter satellite data since 1993 (https://resources.marine.copernicus.eu/). The monthly sea surface height (SSH) anomaly from the geoid is used in this study. Following the definition of DSL according to OMIP (Griffies et al., 2016; Gregory et al., 2019), the area-weighted global mean is removed from the observational data and model data to make them comparable.

For steric sea level anomaly and the depth-integrated density changes, we use the EN4 dataset from the Met Office Hadley Centre, which provides quality-controlled subsurface ocean temperature and salinity profiles based on objective analyses (Good et al., 2013) (https://www.metoffice.gov.uk/hadobs/en4/). The gridded temperature and salinity profiles are used to derive density changes at all available grid cells. We also calculate the density changes solely caused by the thermal expansion (Roquet et al., 2015).

To investigate the possible bias of the wind forcing, we use the Wave and Anemometer-based Sea Surface Wind (WASwind) data (Tokinaga and Xie, 2011). The WASwind provides a global coverage of zonal and meridional wind stress based on wind observation from the International Comprehensive Ocean-Atmosphere Dataset at the monthly frequency with 4° by 4° grid resolution from 1950-2009. Through height-correction for the anemometer-measured winds, the WASwind is not subjected to the spurious upward trend due to increases of anemometer height in the ship-based measurement. The dataset also incorporates the estimated winds from wind wave height. We find this dataset to be suitable for our analysis of the tropical Pacific, and it complements the assessment from Taboada et al. (2018), who focused on upwelling patterns in the global ocean.





## 2.3 Analysis framework

Following Gill and Niller (1973), we separate time tendencies in sea level, $\eta$, into mass and steric contributions (see equation
(2) in (Yin et al., 2010))

$$\frac{\partial \eta(x,y,t)}{\partial t} = \frac{1}{g\,\rho_0}\frac{\partial(p_b(x,y,t) - p_a(x,y,t))}{\partial t} - \frac{1}{\rho_0}\int\limits_{-H(x,y)}^{\eta(x,y,t)}\frac{\partial\rho(x,y,z,t)}{\partial t}\,\mathrm{d}z. \tag{1}$$

In this equation, $g$ is the gravitational acceleration, $p_a$ is the pressure at the ocean surface due to atmospheric mass loading (which is zero in this study when discussing DSL variation in the model), and $p_b$ is the pressure at the ocean bottom. The first term on the right-hand side measures local sea level change due to changes in mass within a seawater column. In the second right-hand side term, $\rho$ is the in-situ seawater density, with density changes integrated over the depth of a water column (from
ocean bottom at $z = -H$ to surface at $z = \eta$ and as normalized by the reference density $\rho_0 = 1025$ kg m$^{-3}$), yielding sea level changes from density (steric) effects. The minus sign on the steric term arises since decreases in density, as from ocean warming, lead to increases in sea level. As noted earlier, the global mean sea level time series is subtracted from all regional sea levels from observations and models to allow for direct comparisons of the resulting DSL.

Based on the 1993-2017 observational data of regional mean sea level from CMEMS and steric sea level from EN4, we find
that the sea level change over this period is dominated by steric effects in the tropical Pacific (defined as the region within the 20°S-20°N zonal band inside the Pacific basin). The regional averaged steric signal accounts for more than 75% of the variance in the total sea level change over the eastern Pacific (180° - eastern boundary) and accounts for more than 85% of the variance in the western Pacific (120°E - 180°) [figure 1]. Particularly, the steric signal within the upper 400 meters accounts for more than 95% of the variance in the total steric signal in both the eastern and western tropical Pacific. This result shows the central
role of density changes in the upper 400 meters, which relates to the thermocline depth changes, in accounting for patterns of sea level variability. The residual of the sea level variance is related to the mass component (equation (1)). These results are consistent with earlier analysis from CORE simulations documented by Griffies et al. (2014).

Density changes in the tropical Pacific are dominated by temperature changes in the upper 400 meters [figure 1]. For the upper 400 meters, such changes can arise from surface heat fluxes as well as lateral and vertical ocean heat transport. The
surface heat flux provides a thermodynamical forcing that directly causes a local thermosteric sea level change (sea level changes due to thermal expansion) through diabatic heating. Surface wind forcing, on the other hand, induces a dynamical effect related to Ekman transport that causes light surface waters to diverge or converge, which in turn modifies sea level. Surface wind stress curl causes a Sverdrup transport associated with both Ekman transport near the surface and geostrophic transport below. The internal wave propogation, like Rossby wave and Kelvin wave, at the seasonal and interannual time scale
also have large effect on the sea level changes in the tropical Pacific. All these effects can have very different contributions to sea level at different time scales and different spatial scales. In this study, we aim to characterize mechanisms that cause sea level variations and trends, and determine reasons for simulation biases.

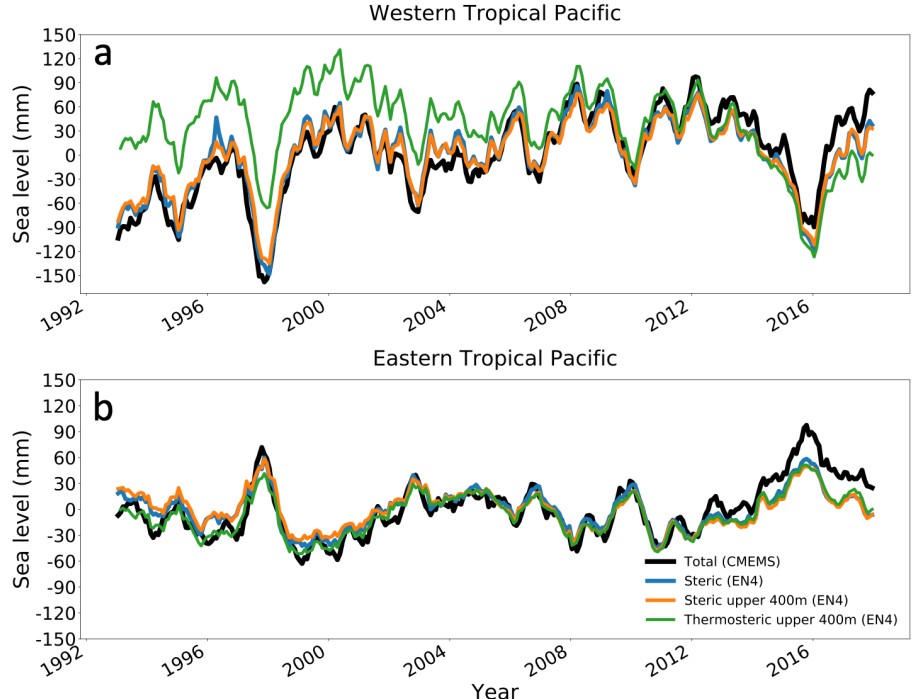

**Figure 1.** Time series of regional mean total sea level from CMEMS (black), steric sea level from EN4 (blue), upper 400 m steric sea level from EN4 (orange), and upper 400 m thermosteric sea level from EN4 (green) in the (a) western tropical Pacific (120°E - 180° & 20°S - 20°N) and (b) eastern tropical Pacific (180° - 60°W & 20°S - 20°N).

## 3    Time Mean Dynamic Sea Level

In this section we focus on the time mean DSL patterns, with the time mean computed over the years 1993-2007 that are
common to both JRA55-do (OMIP-II), CORE (OMIP-I), and observations [figure 2]. This bias pattern is consistent with
Griffies et al. (2014) and Tsujino et al. (2020). The only processing step difference with the previous studies is the Gaussian
smoothing of the observational data. There are two reasons why smoothing is not applicable for this study. One is the larger
Rossby radius of deformation in the tropical region compared to the mid-latitudes. Therefore, the eddies over the tropical
region have a larger spatial scale which can be captured by the model relatively well. The other reason is related to the smaller
spatial scale comparison performed in this study. Smoothing would skew and decrease the amplitude of the smaller scale signal
in the observational data. The following analyses will concentrate on the tropical Pacific region (defined as the region within
the 20°S-20°N zonal band inside the Pacific basin).





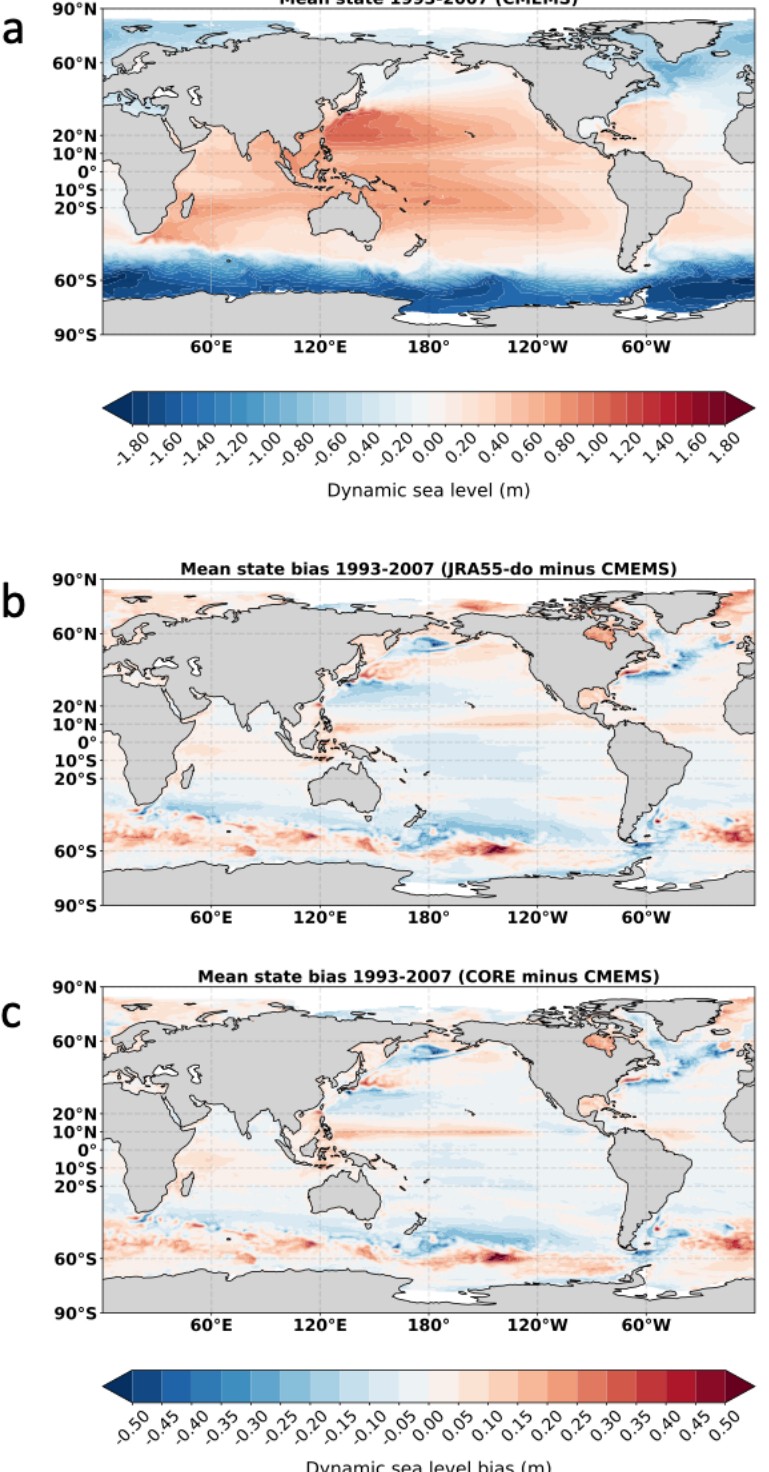

**Figure 2.** (a) The shaded color shows the DSL time mean (1993-2007) from satellite altimeter measurements (CMEMS). The shaded color shows DSL bias in (b) the JRA55-do forced simulation and (c) the CORE forced simulation during the 1993-2007 period.



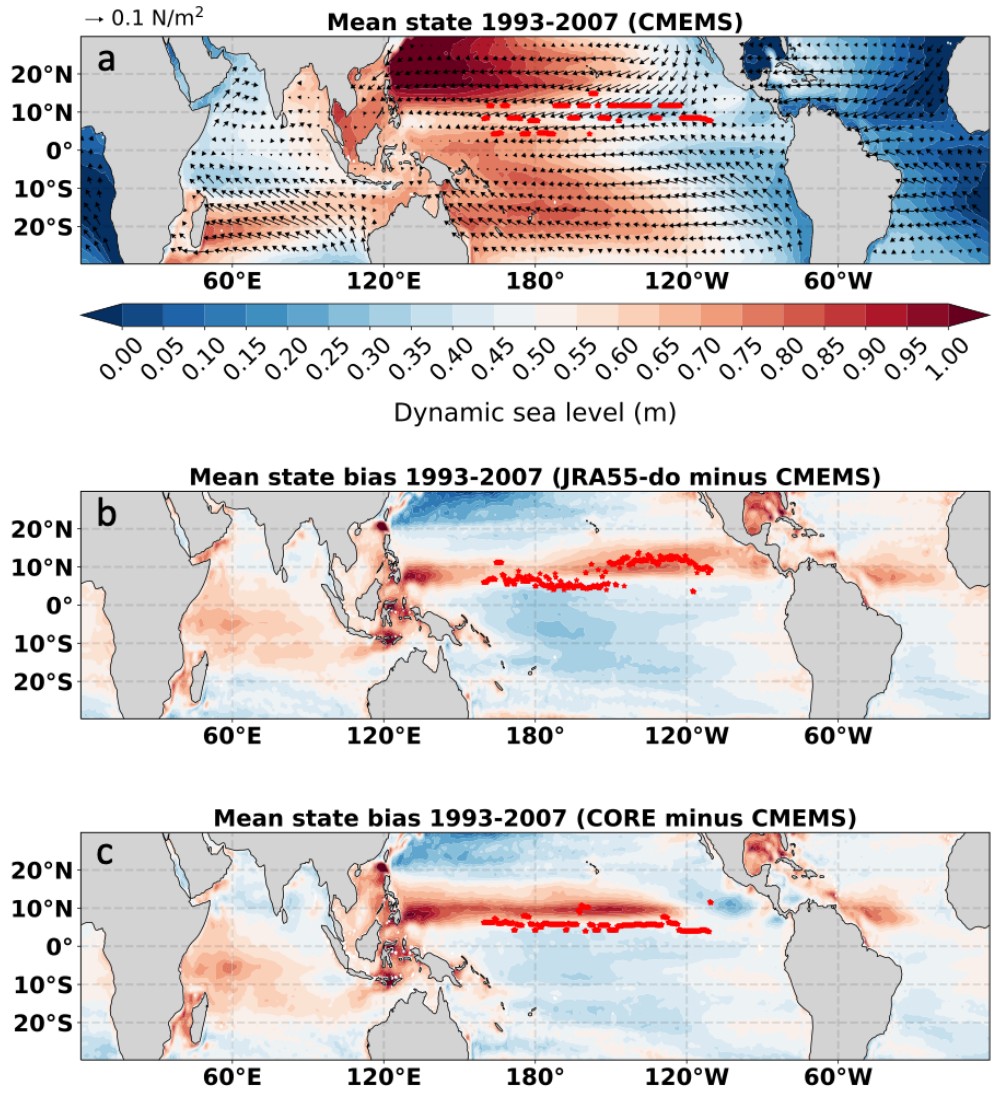

**Figure 3.** (a) The shaded color shows the DSL time mean (1993-2007) from satellite altimeter measurements (CMEMS) and the vector field shows the wind stress time mean from WASwind (see the 0.1 N m$^{-2}$ vector in the upper left for scale). The shaded color shows DSL bias in (b) the JRA55-do forced simulation and (c) the CORE forced simulation during the 1993-2007 period. The red stars indicate the ITCZ location as determined by the maximum wind convergence at each longitude over the tropical Pacific.





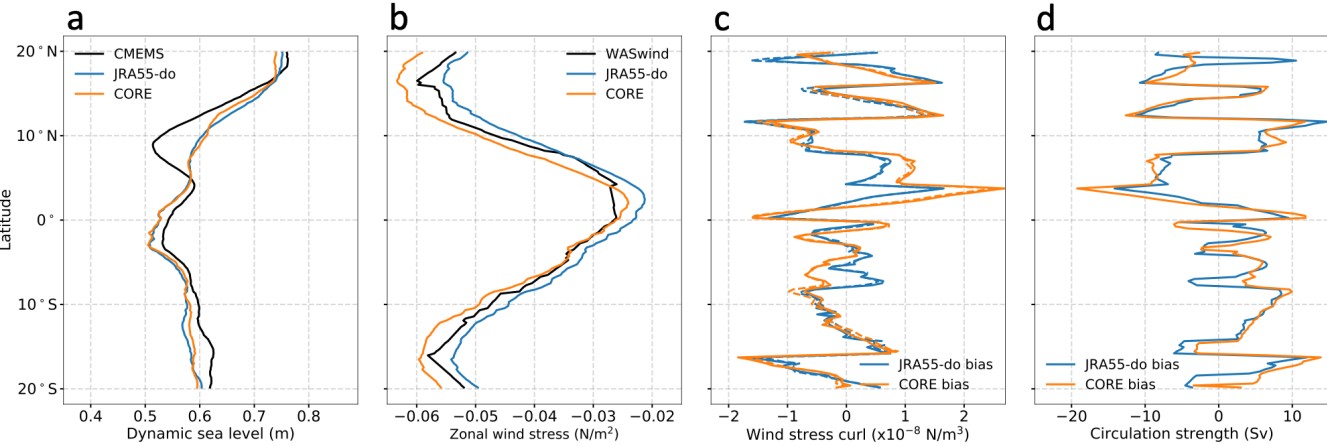

**Figure 4.** Figures show the zonal mean profile of (a) DSL from CMEMS and OMIP simulations (JRA55-do and CORE), (b) zonal wind stress from WASwind and OMIP forcing data (JRA55-do and CORE), (c) wind stress curl bias (solid) and zonal wind induced wind stress curl bias (dashed) for the simulations, and (d) the bias of circulation strength defined as the maximum value of the Sverdrup stream function for the simulations along each latitude over the tropical Pacific.

## 3.1 Role of surface wind stress

Over the tropical Pacific, DSL correlates well with the surface wind stress in the mean state [figure 3a]. Especially in the
Northern Hemisphere, the large scale Ekman transport toward the northwest Pacific, induced by trade winds, results in the
notable DSL ridge related to the subtropical gyre. This DSL ridge creates a down-gradient sea level toward the eastern basin
that generates the eastward pressure gradient force that balances the Coriolis force plus the westward wind stress. To calculate
the mean state bias, we calculate the time-mean DSL during the common period of 1993-2007 for JRA55-do and CORE,
and subtract the CMEMS mean state from the simulations [figure 3b,c]. Both simulations show a negative bias in most of the
tropical Pacific basin except along the zonal band of $10°N$.

Because the bias shows largely zonal structures, we investigate the zonal mean DSL and wind stress in the tropical Pacific.
Figure 4a shows that the DSL in the simulation is generally lower than the observation by about $0.02$ meter except near $10°N$.
The meridional gradient of the zonal mean DSL, which is highly correlated with the narrow zonal currents in the tropical
Pacific, is comparable with the observation in the Southern Hemisphere. The Northern Hemisphere, on the other hand, shows
underestimated DSL gradient in two zonal bands, $9°N$-$20°N$ and $4°N$-$9°N$ due to the lack of DSL trough around $9°N$ and DSL
ridge at $4°N$. Between $9°N$-$20°N$, the down-gradient DSL toward the equator is associated with the strength of north equatorial
current (NEC). Between $4°N$-$9°N$, the up-gradient DSL toward the equator is associated with the strength of north equatorial
counter-current (NECC). Therefore, the smaller or missing DSL gradient in both simulations could lead to underestimated
NEC and NECC.





For the mean state of surface wind forcing, we compare the wind stress curl and the associated circulation strength (defined as the maximum value of the Sverdrup stream function along each latitude) between the forcing dataset and the observation from WASwind. The Sverdrup stream function is defined as

$$\Psi = - \int\limits_{EB}^{x} \frac{\nabla \times \boldsymbol{\tau}}{\beta \rho_0} \mathrm{d}x, \tag{2}$$

where $EB$ is the eastern boundaries of the tropical Pacific at each latitude, $\boldsymbol{\tau}$ is the wind stress vector, $\beta = df/dy$ is the
meridional derivative of the Coriolis parameter, $f$, and $\rho_0 = 1025$ kg m$^{-3}$ is the reference density. Figure 4b-d show the zonal mean of zonal wind stress ($\tau_x$), wind stress curl ($\nabla \times \boldsymbol{\tau}$) bias, and circulation strength bias, respectively. The wind stress curl bias is dominated by the zonal wind shear bias [figure 4c]. Although the zonal wind stress is generally similar between observations and simulations [figure 4b], the wind stress curl bias is sensitive to even small differences. The wind stress curl bias shows a large positive bias at around 4°N and a negative bias at around 9°N [figure 4c]. The JRA55-do and CORE do not
exactly capture some subtle features in the observation, such as the sharp zonal wind shear across the Intertropical Convergence Zone (ITCZ) and the comparatively uniform zonal wind stress in 0-5°N latitudes. This difference creates a negative bias in Sverdrup stream function (counter-clockwise circulation bias) at 4°N and positive bias (clockwise circulation bias) at 9°N [figure 4d]. Since the geostrophic flow components dominate the Sverdrup stream function, the counter-clockwise circulation bias represents a negative DSL bias at the circulation center, and the clockwise circulation bias represents a positive DSL
bias at the circulation center. These opposite DSL biases induced by the wind stress flatten the DSL ridge and trough in this region, thus decreasing the sea level gradient in both zonal bands. This analysis highlights the importance of an accurate representation of the zonal wind stress shear for simulating the DSL trough and the associated zonal currents in the 4°N-9°N band of the tropical Pacific.

### 3.2 The role of zonal currents in the ocean heat budget

To analyze the impact of zonal currents on heat, we consider two boxes within the tropical Pacific and assess the mean heat advection and volume transport from individual currents. For this purpose, we define a western box (Wbox with boundaries 120°E - 180°; 20°S - 20°N; 0-400m) and an eastern box (Ebox with boundaries 180° - eastern boundary; 20°S - 20°N; 0-400m) as shown in figure 5, and express the heat budget as

$$\iiint\limits_{V} \frac{\partial Q}{\partial t} dV = F_{srf} - \rho_0 C_p \iint\limits_{S} (\boldsymbol{u} \cdot \hat{\boldsymbol{n}})(T - T_m) dS + < \text{residual} > . \tag{3}$$

In this equation, $\iiint_{V} \frac{\partial Q}{\partial t} dV$ is the heat content changes in a box, which is related to the thermosteric sea level changes, and $F_{srf}$ is the area integrated net surface heat flux. $T_m$ represents the volume mean temperature inside a box. For Wbox, $T_m = T_{mw}$ and for Ebox, $T_m = T_{me}$. We calculate the heat advective transport term following the analysis of Lee et al. (2004) and Ray et al. (2018). The residual term accounts for transport from small-scale processes including vertical mixing, sub-grid scale processes, with these processes not of concern here and are small compared to the role of currents.



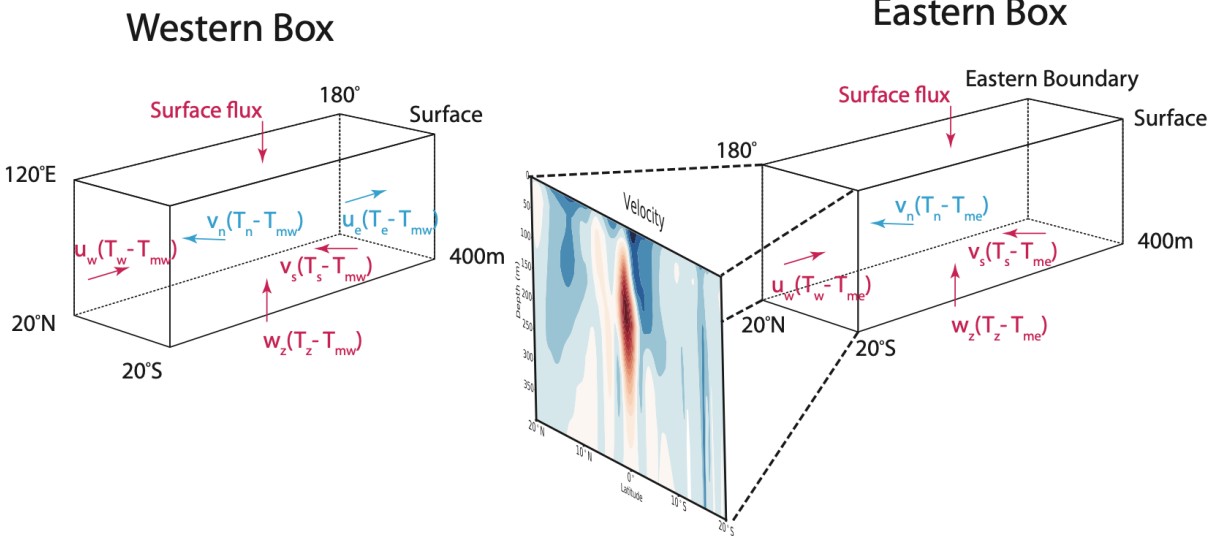

**Figure 5.** The two-box regions for the heat budget analysis, with the western box (Wbox) and the eastern box (Ebox) shown over the tropical Pacific. The western box is from $120°$E - $180°$, $20°$S - $20°$N, and 0-400 m, and the eastern box is from $180°$ - eastern boundary, $20°$S - $20°$N, and 0-400 m. All variables are a function of space and time. $T$ is potential temperature; $u$,$v$,and $w$ are the velocity components in the $\hat{x}$, $\hat{y}$, and $\hat{z}$ directions, respectively. Subscripts $w$, $e$, $s$, $n$, and $z$ represent western, eastern, southern, northern, and bottom boundary of the boxes, respectively. $T_{mw}$ and $T_{me}$ represent the volume mean temperature in the Wbox and the Ebox, respectively. The velocity field transect across $180°$ between Wbox and Ebox is shown by the shaded field, where positive value represents eastward flow. A detailed defined current systems based on the velocity field are shown in figure 6d.

The heat advection across the $180°$ transect dominates the heat content changes (thermosteric sea level changes [figure 1]) in the Wbox [figure 6], whereas a compensation between surface heat flux and heat advection across the $180°$ transect is important in the Ebox. For the Ebox, the compensation between $180°$ heat advection and surface heat flux shows the eastern tropical Pacific as an important region for heat to enter the ocean and then to participate in the global energy cycle, which is consistent with the finding of diabatic heating at the surface controlling the heat movement over the ocean (Holmes et al.,

2019). Despite the large net surface heat flux in the Ebox, the heat is not stored in the eastern tropical Pacific but flushed to the western tropical Pacific. We find that, with larger net heat flux over the Ebox, the heat advection across the $180°$ transect is larger in CORE than JRA55-do. For the Wbox, the main heat loss occurs at the $120°$E transect which advects the heat out with the magnitude a little less than half of the heat coming from the eastern tropical Pacific. Heat loss also occurs across the other three boundaries of the Wbox but considerably smaller than across the $120°$E transect. Besides some small differences

in surface heat flux and the heat advection at $180°$ transect, we do not see significant differences in the mean state heat budget between the simulations forced by CORE and JRA55-do.

  Due to the importance of zonal heat transport between the two boxes, we look into the dominant currents and their relative contributions to the $180°$ transect [figure 6b,c]. We define the current based on the time-mean of the zonal velocity [figure 6d].



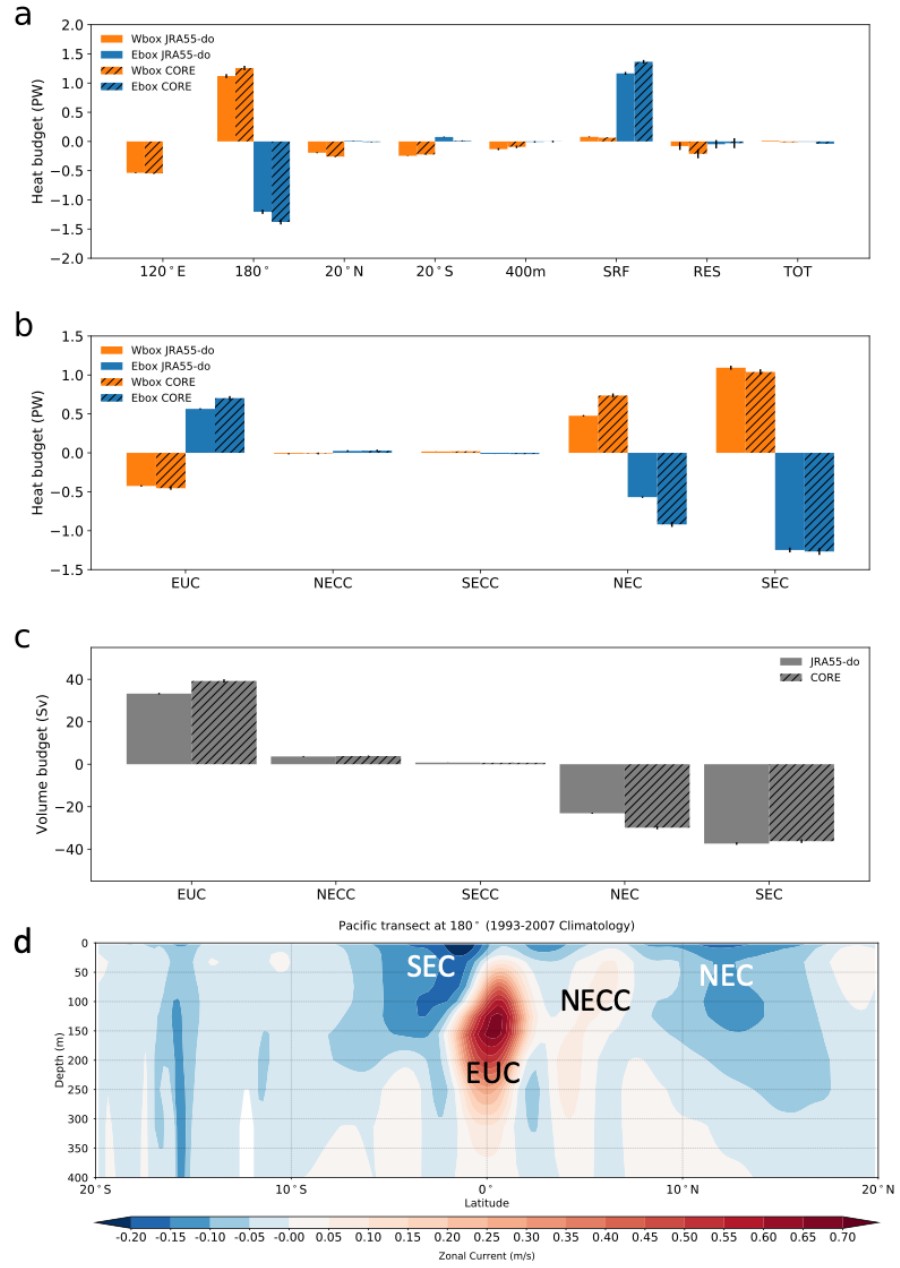

**Figure 6.** The 1993-2007 time mean (a) heat contribution from advection across different transects (120°E, 180°, 20°S, 20°N, 400m),
surface flux (SRF), and the residual processes calculated as the difference between the total heat content tendency in the box (TOT) and the
sum of SRF and heat advection from all boundaries. The filled yellow and blue bar represents Wbox and Ebox, respectively. The JRA55-do
(no stripe) and the CORE (with stripes) forced simulation are shown side by side for comparison. The mean state of individual current system
contributing to the (b) heat budget through heat advection and (c) the associated volume transport across 180° transect. The error bar shows
the 99% confidence interval of the mean value. (d) The mean zonal velocity from the JRA55-do simulation at 180° transect in the tropical
Pacific where positive means eastward velocity. The text shows each defined current system.







**Figure 7.** The 1993-2007 time mean net surface heat flux in JRA55-do (top), CORE (middle), and CORE minus JRA55-do (bottom). Positive value means energy flux into the ocean. Note the different scales of the two color bars.

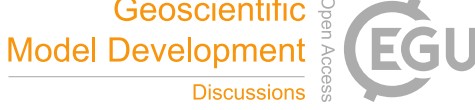

The south equatorial current (SEC) is defined by the westward mean zonal velocity within 20°S to 5°N zonal band and above

400 meters. Both CORE and JRA55-do forced simulation show that the SEC makes the largest contribution to the heat content changes in Wbox and Ebox with similar magnitude. The north equatorial current (NEC) which is defined by the westward mean zonal velocity within 5°N- 20°N zonal band and above 400 meters shows the second-largest contribution. The heat advection of NEC in CORE is around 50% larger than JRA55-do. This difference in heat advection is a result of the larger mean zonal current speed in the CORE-forced simulation [figure 6c]. Interestingly, we also see a higher energy input in the northern part of

Ebox [figure 7] in CORE, which reiterates the concept that the energy input in the Ebox is always flushed toward the Wbox in the simulations. The equatorial undercurrent (EUC) which is defined by the eastward mean zonal velocity within 2.5°S-2.5°N zonal band largely compensates the westward heat advection by the SEC and the NEC in both JRA55-do and CORE forcing. Both the north equatorial counter-current (NECC) defined by the eastward mean zonal velocity within 2.5°N to 10°N zonal band and south equatorial counter-current (SECC) defined by the eastward mean zonal velocity within 2.5°S to 10°S zonal

band show little contribution to heat content change for both boxes.

**Table 1.** Comparison between observation and JRA55-do/CORE simulations for various tropical Pacific currents.

| Currents | characteristics | JRA55-do | CORE | Observations |
|---|---|---|---|---|
| EUC (2.5°S-2.5°N) | max speed | 1.2 m s$^{-1}$ | 1.2 m s$^{-1}$ | 1.1 m s$^{-1}$ (Johnson et al., 2002) |
| | max speed depth | 100 m | 100 m | 86 m (Johnson et al., 2002) |
| | max speed longitude | 138°W | 129°W | 130°W (Johnson et al., 2002) |
| | meridional extent at 180° | 2°S-2°N | 2°S-3°N | 2°S-2°N (Johnson et al., 2002) |
| SEC (20°S-2.5°N) | max speed at 180° | 0.2 m s$^{-1}$ | 0.2 m s$^{-1}$ | 0.2 m s$^{-1}$ (Johnson et al., 2002) |
| NECC (10°S-2.5°S) | max speed at 180° | 0.1 m s$^{-1}$ | 0.1 m s$^{-1}$ | 0.4 m s$^{-1}$ (Johnson et al., 2002) |
| NEC (2.5°N-20°N) | volume transport at 180° | 23 Sv | 29 Sv | 40 Sv (Zhang et al., 2017) |

To check for current biases, we compare the speed and volume transport of each current system in JRA55-do and CORE with the available observational data [table 1]. The EUC is well-represented with simulated maximum value of 1.2 m s$^{-1}$ at approximately 100 m depth at 138°W in JRA55-do and 1.2 m s$^{-1}$ at approximately 100 m depth at 129°W in CORE along the equator, which compares to an observed value of 1.1 m s$^{-1}$ at approximately 86 m at 130°W from Johnson et al. (2002).

The meridional extension of the EUC is also well simulated when comparing to the observed current confined between 2°S and 2°N at 180° in JRA55-do and 2°S and 3°N in CORE which skews more toward the Northern Hemisphere (Johnson et al., 2002). The SEC at 180° shows good agreement in strength with the observed maximum value of 0.2 m s$^{-1}$ for both JRA55-do and CORE (Johnson et al., 2002). Due to the underestimated DSL gradient in the Northern Hemisphere and the missing DSL trough, weaker NEC and NECC in JRA55-do and CORE are expected. The observed NECC strength is 0.4 m s$^{-1}$ while the

simulations show a much weaker value of 0.1 m s$^{-1}$. The weak NECC bias exists in the JRA55-do forced simulation as well as the CORE forced simulation, with Tseng et al. (2016) attributing the CORE simulation biases to an inaccurate wind stress





representation. Unfortunately, the improved surface wind forcing from JRA55-do did not resolve the underestimation of the NECC. We hypothesize that the key reason for the weak NECC is due to both the underestimated zonal wind stress in JRA55-do *and* a flattening of the DSL trough due to the wind stress curl bias in the northern tropical Pacific found in both CORE and

JRA55-do [figure 4b,c]. The NEC is also underestimated due to the flattening of the DSL trough for both simulations. Namely, the NEC transport in the JRA55-do forced simulation is 23 Sv and 29 Sv in CORE, whereas the Argo derived value is near 40 Sv (Zhang et al., 2017).

When we compare NECC and NEC in both model simulations, the heat advection between Wbox and Ebox from NEC is significantly larger than NECC. Therefore, the Northern Hemisphere heat content changes in Ebox and Wbox are dominated by

the NEC. Heat content changes are related to thermosteric sea level changes, so that a stronger NEC heat advection can explain the stronger east-west gradient in the DSL trend found in the Northern Hemisphere in CORE relative to JRA55-do [figure 8b,c]. However, the weak NEC bias and difference between CORE and JRA55-do cannot explain the consistent underestimation of the DSL trend along the 10°N-20°N zonal band in both the CORE and JRA55-do forced simulations [figure 9a,b].

## 4   Dynamic Sea Level Trend

Over the 1993-2012 period, satellite altimetry has shown a significant sea level rise which is associated to the warming trend in the western tropical Pacific [figure 8]. The trend has been extensively studied due to its significant asymmetric zonal pattern over the tropical Pacific which is related to global warming rates (Peyser et al., 2016; Merrifield, 2011). Though the asymmetry in the DSL trend is significant in the observations, ocean model simulations forced by CORE have failed to represent this DSL trend pattern (Griffies et al., 2014). Our model simulations consistently underestimate the DSL trend along the 10°N-20°N

zonal band even with the updated surface forcing from JRA55-do [figure 9]. The reason for this simulated DSL trend bias has not been discussed in the literature.

Figures 9a,b show the DSL trend and bias by subtracting the CMEMS DSL trend from the simulations forced by CORE and JRA55-do. The trend bias, though still present, is significantly reduced with JRA55-do. The zonal mean DSL trend between 10°N-20°N is underestimated by 1 mm yr$^{-1}$ on average in JRA55-do and much larger bias of 4 mm yr$^{-1}$ underestimation

with CORE [figure 10a]. By substracting the JRA55-do simulation from the CORE simulation, we also find that the difference between CORE and JRA55-do are similar to the CORE bias (CORE minus CMEMS) [figure 9c]. This result shows the bias reduction in JRA55-do is significant and has a trend better matching the CMEMS DSL trend.

Furthermore, we extend the trend calculation to 50 years (1958-2007), and calculate the trend difference between the JRA55-do and CORE simulations [figure 9d]. The band structured DSL trend difference persists for the longer time period (1958-2007)

[figure 9d]. The zonal mean of DSL trend in the tropical Pacific also shows the underestimated DSL trend between 5°N-20°N [figure 10b].



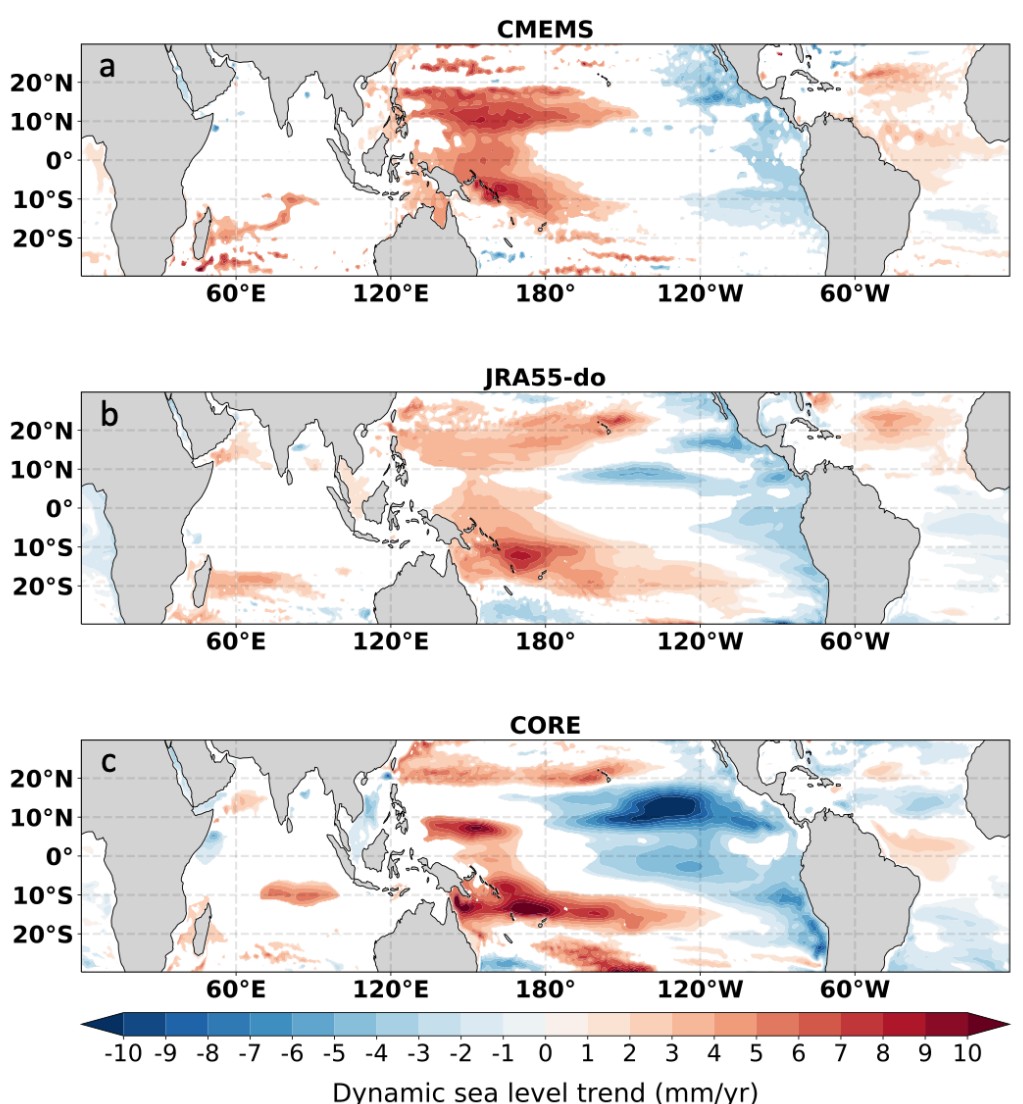

**Figure 8.** The linear DSL trend during the 1993-2007 period derived from (a) satellite altimeter observation (CMEMS), (b) JRA55-do forced simulation, and (c) CORE forced simulation. The shaded color shows the trend that is statistically significant with 99% confidence.





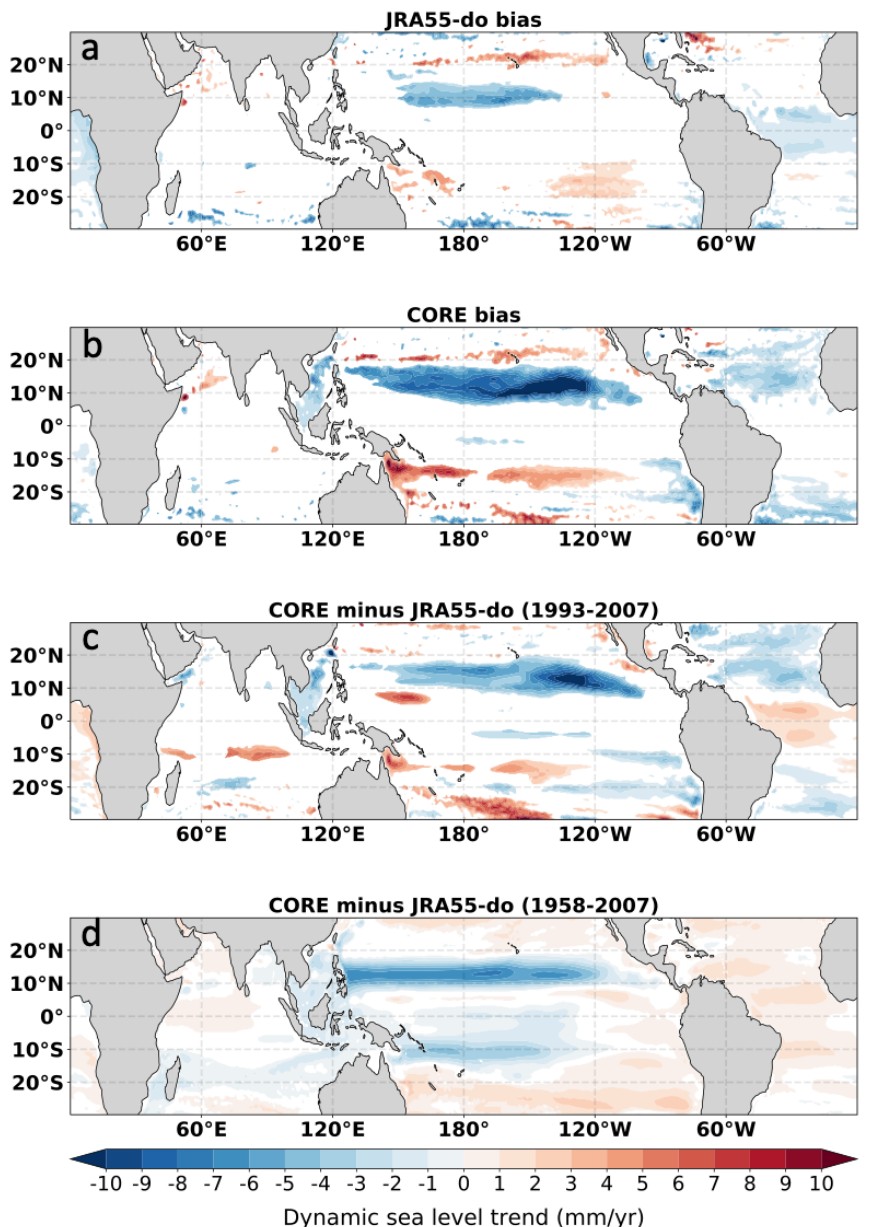

**Figure 9.** The DSL trend bias (CMEMS derived DSL subtracted from the simulations) during the 1993-2007 period common to CORE and JRA55-do. Panel (a): the JRA55-do forced simulation. Panel (b): the CORE forced simulation. DSL trend difference determined by subtracting JRA55-do from CORE during 1993-2007 in panel (c) and during 1958-2007 in panel (d). The shaded colors show the trend biases or differences that are statistically significant with 99% confidence.



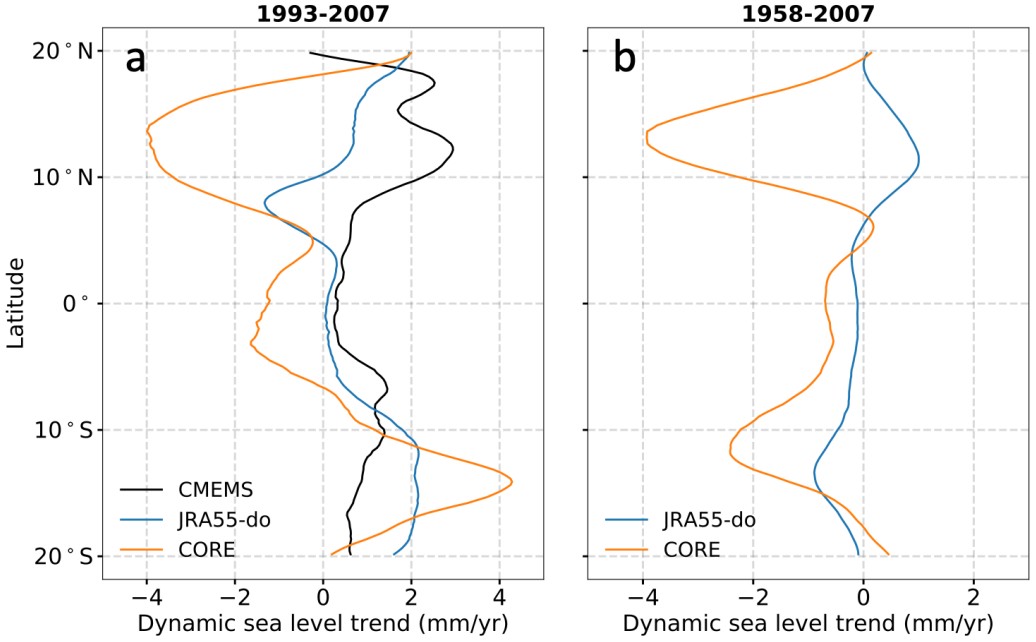

**Figure 10.** The zonal mean DSL trend from observations and the JRA55-do and CORE simulations during (a) 1993-2007 and simulations during (b) 1958-2007 over the tropical Pacific basin.

## 4.1 Ekman layer response

To help reveal mechanisms for the underestimation of the DSL trend, we investigate the wind stress forcing in CORE and JRA55-do and compare with the WASwind observational product. We calculate the wind stress trend bias by subtracting the
WASwind wind stress trend from CORE and JRA55-do. During 1993-2007, there is no statistically significant trend bias in wind stress that can help explain the DSL trend bias. For trend significance test, the null hypothesis in the statistical test is zero long-term trend with degree of freedom equal to the number of monthly data minus two due to the intercept and slope determined during the linear fit. The large variability of wind stress and the small number of samples during this period result in an inconclusive statistical test. However, with the increased number of samples during a longer period (1958-2007), a
statistically significant zonal wind bias in JRA55-do and CORE can be found [figure 11].

The most obvious bias is the excessive westerly wind trend in CORE and JRA55-do than WASwind located at the central and eastern Pacific, with the trend bias particularly large in CORE [figure 11c]. A westerly trend in CORE extends from 10°S to 15°N can also be seen in the zonal average over the tropical Pacific [figure 12]. The westerly trend is significantly reduced in the JRA55-do. On top of the westerly trend bias, there is an easterly trend bias in the 15°N-25°N zonal band for both JRA55-
do and CORE [figure 11b,c]. JRA55-do has an easterly trend bias only west of 150°W while CORE extends across the entire Pacific basin. The existence of both the westerly and easterly trend biases in CORE generates a strong positive wind stress curl trend in the 8°N-20°N zonal band [figure 12b]. The JRA55-do simulation also has a positive wind stress curl trend bias in the



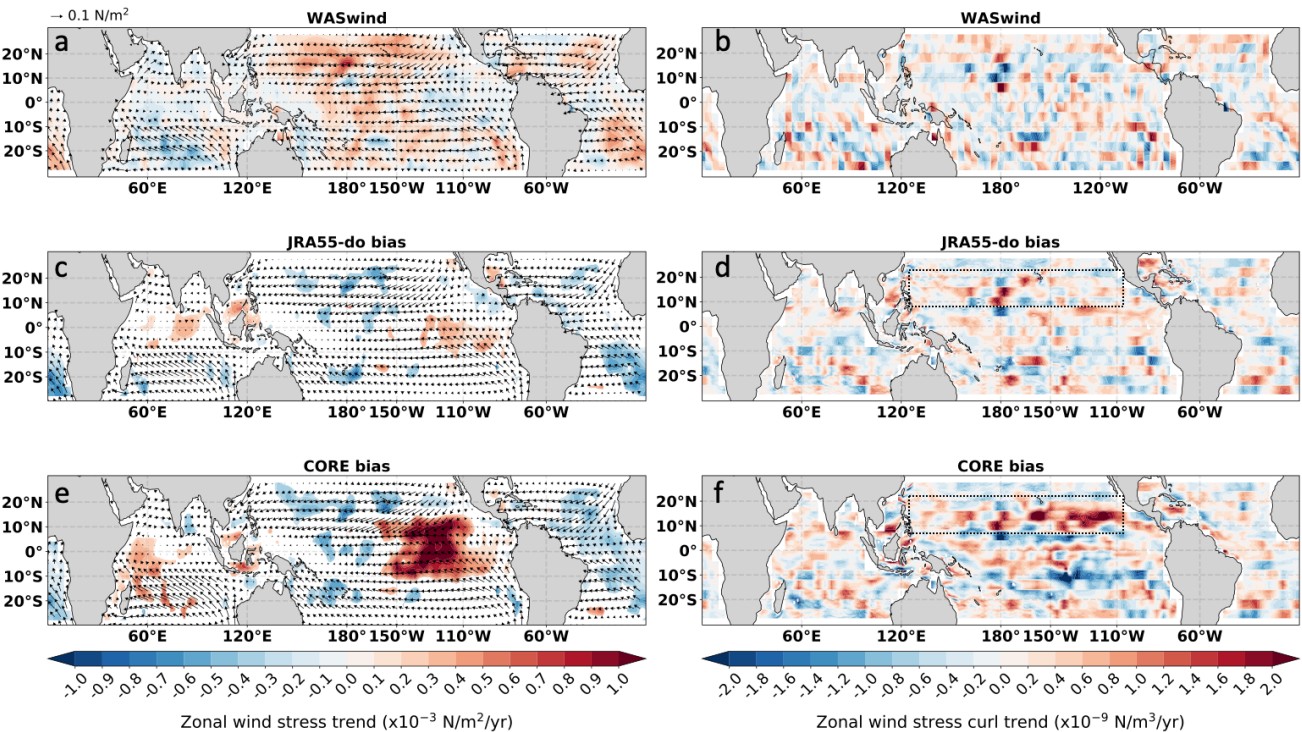

**Figure 11.** (a) The zonal wind stress trend (shading) and (b) wind stress curl trend (shading) during 1958-2007 in the observational data (WASwind). (c) The zonal wind stress trend bias (shading) and (d) wind stress curl trend bias (shading) over the same period in JRA55-do forcing data. The same as (c,d) but with (e,f) CORE forcing data. The shading in (c,e) shows the trend is statistically significant with 99% confidence. The vector field shows the mean state of wind stress during the same period for (a) WASwind, (c) JRA55-do, and (e) CORE. The black dashed box is the region of strong negative DSL bias in both simulations.

same zonal band but is roughly three times smaller than the CORE simulation [figure 12]. The smaller wind stress curl trend bias with JRA55-do is mainly due to the missing westerly trend bias in the 10°S to 10°N region.

The westerly trend bias in CORE forcing data is a result of the multiplicative factor ($R_s$) applied to the vector winds in the reanalysis data by Large and Yeager (2009). The multiplicative factor is determined by the five-year mean ratio between reanalysis data and observational data. Since the factor is designed to make the mean state of wind amplitude in reanalysis data better fit the observation, applying the same factor to the entire time series could result in biases across different time scales. In this case, the factor designed to correct the mean state causes an overestimation of the trend in the westerly wind in the eastern

tropical Pacific where the factor has the highest value in the tropical Pacific [see figure 2a in Large and Yeager (2009)]. The modified multiplicative factor (called offsetting factor) in Tsujino et al. (2018) has a better adjustment without introducing the westerly trend bias. On the other hand, the easterly bias in the 15°N-25°N zonal band exists in both the JRA55-do and CORE forcing data.





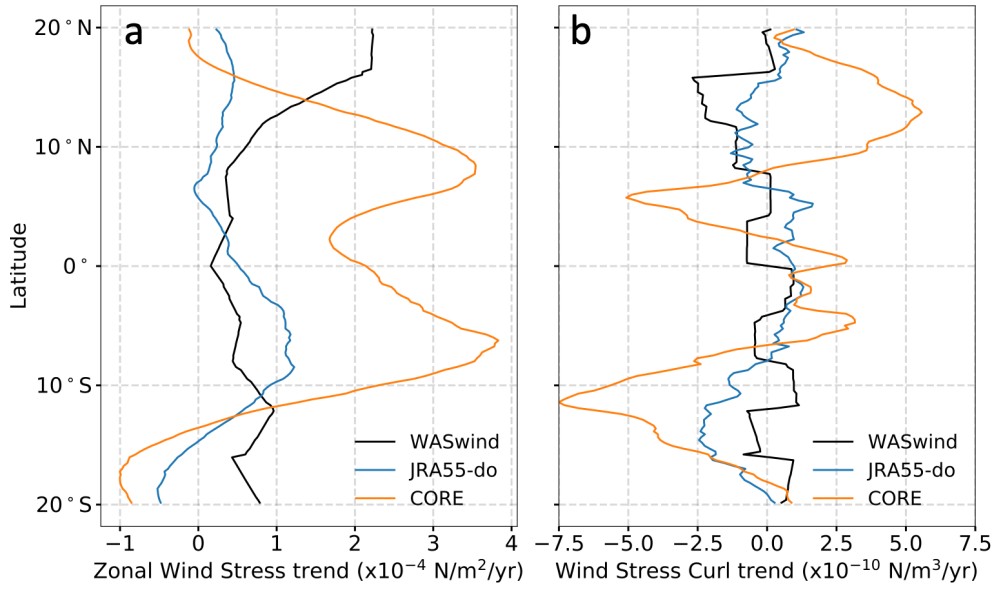

**Figure 12.** The zonal mean during the 1958-2007 period for (a) zonal wind stress trend and (b) wind stress curl trend over the tropical Pacific basin.

We conclude from this analysis that the positive wind stress curl trend bias west of 150°W, found in both CORE and JRA55-do, is mainly due to the easterly wind stress trend bias. Additionally, the positive wind stress curl trend bias east of 150°W in the CORE forcing is due to the combined effect of easterly and westerly trend biases. The positive wind stress curl trend bias in the zonal band creates an artificial Ekman suction that causes the DSL trend in the simulations to be biased low.

### 4.2 Barotropic response

We now examine how the barotropic geostrophic response affects the DSL trend. From the trend bias in the Sverdrup stream function, we can see how the geostrophic flow trend bias can cause the DSL trend bias. The Sverdrup stream function ($\Psi$) is calculated following equation 2. The CORE simulation shows a negative stream function trend bias in the 10°N-20°N zonal band, whereas the JRA55-do trend bias is relatively small [figure 13a,b]. The negative trend bias which corresponds to a trend of counter-clockwise geostrophic current is related to the negative DSL trend bias across the Pacific basin.

To further investigate the dominant causes, we perform the same Sverdrup stream function calculation starting from the eastern boundary. However, in this calculation we stop at 150°W where the largest westerly wind trend bias is located, and keep the remaining basin as zeros until reaching the western boundary [figure 13c,d]. This calculation helps to identify the contribution from the westerly trend bias in the eastern tropical Pacific. In particular, it reveals the importance of the westerly trend bias in the eastern tropical Pacific in driving the large scale counter-clockwise circulation trend bias, while the easterly wind bias along the 10°N-20°N latitude band west of 150°W strengthens the circulation trend bias in the CORE simulation. The Southern Hemisphere also shows the same negative DSL trend bias that can be explained by the same Ekman suction and





the large scale Sverdrup balance during 1958-2007. This analysis shows the significant effect of zonal wind stress trend biases near the equator on the off-equatorial DSL trend biases from geostrophic balance and Ekman suction.

The large interannual variability in DSL can affect the trend bias estimates over a shorter period. Based on the improved forcing from JRA55-do, we calculate the DSL trend during 1993-2017 and compared with the available observational data. 290 Despite the reduced trend bias when comparing to 1993-2007, figure 14 still shows the trend bias in the 10°N-20°N zonal band, which also shows the existence of the DSL trend bias is not due solely to interannual variability over a short time period.

## 5   Dynamic Sea Level Seasonal Variability

Seasonal variability of DSL in the tropical Pacific is significant, especially over the Northern Hemisphere between 2°N to 10 °N. During the December, January, and February (DJF) mean, we can find a clear zonal dipole structure between the western 295 and eastern Pacific in the 2°N to 10 °N zonal band [figure 15c,d]. The dipole pattern completely reverses during the June, July, and August (JJA) mean, which is strongly related to the location of the positive wind stress curl [figure 15a,b]. The wind change associated with the ITCZ seasonal migration in the eastern tropical Pacific is the main cause of the seasonal variability in this zonal band. For the 2°S to 10 °S zonal band, we see the same signal but significantly weaker than the Northern Hemisphere counterpart. As for latitude poleward of 10°N, the seasonal variation synchronizes across the Pacific basin similar to the mid- 300 latitude response where the surface net heat flux dominates the changes of seasonal DSL.

### 5.1   Rossby wave propagation

The seasonal variation of DSL in the 2°N to 10 °N zonal band is a result of the Rossby wave propagation that is generated in the eastern and central tropical Pacific by the local wind stress curl (McPhaden et al., 1988). Figure 15 shows the spatial pattern of wind stress curl anomaly and the DSL anomaly in the eastern tropical Pacific between 2°N to 10 °N. 305 The Hovmöller diagram of the meridional mean from 2°N to 10 °N shows the propagation of this seasonal DSL signal and its relation with the Ekman pumping derived from wind stress curl [figure 16]. The DSL signal propagation speed matches the theoretical value of the first baroclinic Rossby wave of $0.3 \text{ m s}^{-1}$ (Knauss and Garfield, 2016). A dipole structure appears across the zonal band when the previously generated signal reaches the central Pacific while a newly generated opposite signal starts in the eastern tropical Pacific. The zonal dipole DSL does not reverse the absolute DSL gradient since the DSL difference 310 between the east and west is 0.6 m in the mean state while the Rossby wave created a east-west dipole that has an amplitude of only 0.1 m.

The CORE simulation also shows a larger seasonal DSL bias than JRA55-do. The larger bias in CORE is mainly due to the wrong timing of the wind stress curl in the eastern tropical Pacific. Generally, the seasonal amplitude of DSL forced by CORE is closer to the observed amplitude than JRA55-do. However, the timing of the DSL signal is off by three months, especially 315 in the initiation region of 90°W [figure 16]. The positive DSL anomaly only starts in May for CORE while JRA55-do and observation show positive anomaly as early as February along 90°W. The DSL lag results in a much longer and larger negative DSL anomaly signal around 100°W in the CORE simulation.



**Figure 13.** The Sverdrup stream function trend bias during 1958-2007 period in (a) JRA55-do and (b) CORE, which is derived from integrating through the whole Pacific basin as is. By changing the value to zero west of 150°W to the western boundary, the Sverdrup stream function trend bias is calculated during the same period in (c) JRA55-do and (d) CORE.





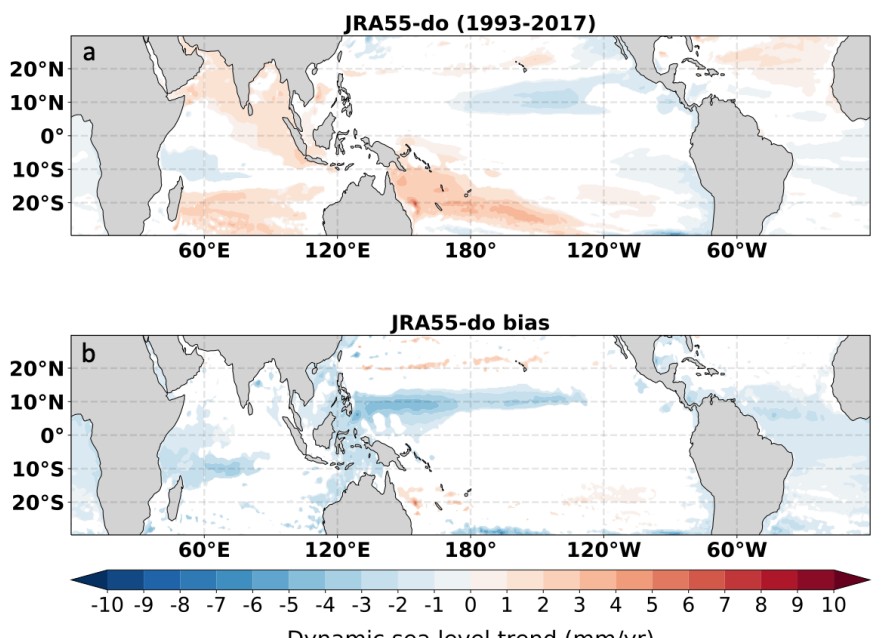

**Figure 14.** The DSL (a) trend and (b) trend bias during 1993-2017 with 99% confidence (shading) in the JRA55-do forced simulation.

The missed timing in CORE is related to the weaker Ekman pumping/suction east of 90°W throughout the year, thus causing significant DSL biases when comparing to observations. The timing of the JRA55-do simulation is relatively close to observations but with underestimated amplitudes in both Ekman pumping and DSL. This analysis shows the importance of resolving dynamics near the ocean boundary as they can strongly affect the basin-scale DSL variation on seasonal time scales.

Outside the initiation region, both forcing data show weaker Ekman suction during JJA and weaker Ekman pumping during DJF between 150°W and 180° when compared to observations [figure 16b,d]. The weaker forcing does not affect the propagation of the DSL signal, but it causes the DSL signal in both simulations to be biased low west of 150°W due to the lack of continuous external forcing.

The vertical Ekman-induced velocity in the Ekman layer is given by

$$W_E = \mathbf{k} \cdot \nabla \times (\frac{\boldsymbol{\tau}}{\rho_0 \, \mathrm{f}}) = \frac{1}{\rho_0 \, \mathrm{f}}(\frac{\partial \tau_y}{\partial x} - \frac{\partial \tau_x}{\partial y} + \frac{\tau_x}{\mathrm{f}}\beta). \tag{4}$$

In the tropics, a vertical velocity in the Ekman layer can be generated with zero wind stress curl but strong zonal wind due to the large $\beta$ effect. Therefore, the weaker Ekman suction during JJA and weaker pumping during DJF could be related to the underestimated wind stress curl or zonal wind stress between 150°W and 180°. The wind stress curl does not show bias as Ekman pumping/suction bias [figure 17f,h]. The zonal wind stress, on the other hand, shows the negative bias (easterly bias) during JJA and positive bias (westerly) during DJF between 150°W and 180° in both JRA55-do and CORE, which is similar to the bias in Ekman pumping/suction [figure 17b,d]. This result indicates that the westerly (during JJA) and easterly (during



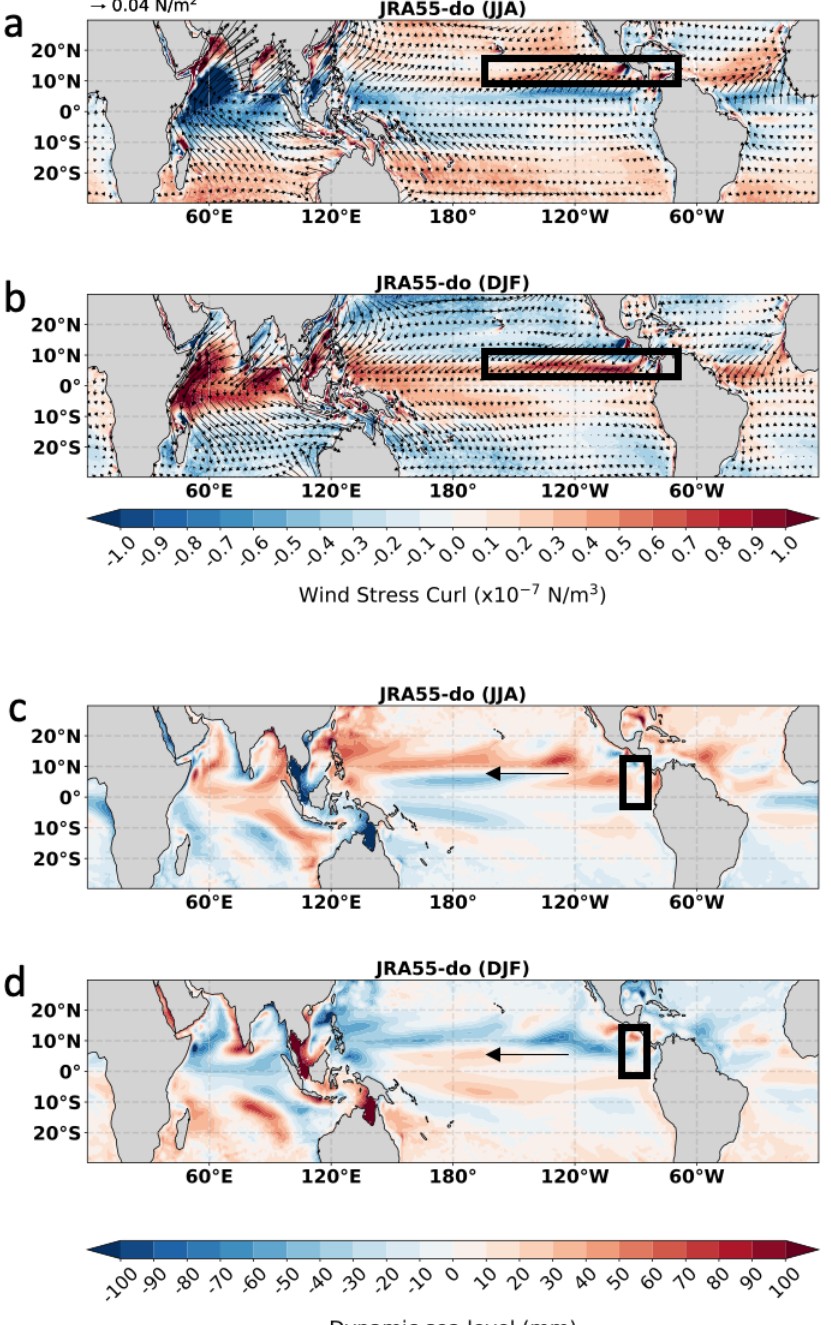

**Figure 15.** Monthly climatology (1993-2007) of (a) June, July, August (JJA) and (b) December, January, February (DJF) in wind stress (vector) and wind stress curl (shading) from JRA55-do forcing data with mean state removed. The corresponding DSL of (c) JJA mean and (d) DJF mean from the JRA55-do forced simulation. The black box in (a,b) is the positive wind stress curl location which is associated to the ITCZ location in the eastern tropical Pacific. The black arrow in (c,d) show the Rossby wave propagation direction with black box showing the initiation region in the eastern tropical Pacific.



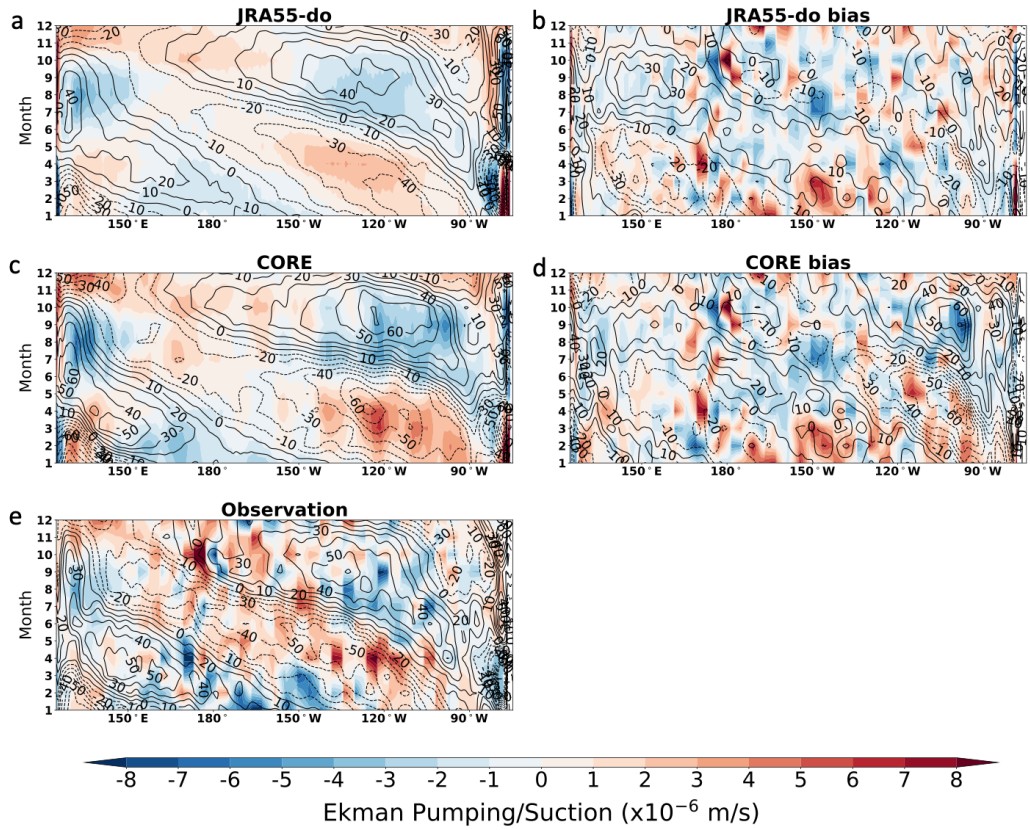

**Figure 16.** Hovmöller diagram of monthly climatology with mean state removed showing the meridional mean (2°N to 10°N) DSL (contour in mm) and derived Ekman pumping/suction (shading) in (a) JRA55-do forced simulation and (b) the associated bias, and in (c) CORE forced simulation and (d) the associated bias. (e) The DSL (contour in mm) from CMEMS and derived Ekman pumping/suction (shading) from WASwind.

DJF) winds between 150°W and 180° are not strong enough in both JRA55-do and CORE, which leads to the underestimated
Ekman suction and pumping, respectively.

The other interesting fact is the compensating effect between zonal wind stress and wind stress curl on the Ekman mechanism in this zonal band. In other words, the first two terms on the right hand side of equation 4 which represents the wind stress curl induced Ekman effect compensates with the last term which represents the zonal wind stress induced Ekman effect. Figure 17 quantitatively visualizes this compensating effect perfectly, which shows the zonal wind related Ekman contribution and
wind stress curl related Ekman contribution. During JJA, the westerly wind dominates the zonal band due to cross-equatorial flow from the Southern Hemisphere, converging toward the ITCZ located in the Northern Hemisphere [figure 15a]. The cross-equatorial flow, at the same time, provides a negative wind stress curl with Ekman pumping that compensates the effect of westerly wind on generating the Ekman suction [figure 17a,e]. During DJF, the easterly component is generated due to the dominant northeasterly wind converging toward the ITCZ with little zonal wind south of the ITCZ [figure 15b]. The difference





**Figure 17.** Hovmöller diagram of monthly climatology with mean state removed showing the meridional mean (2°N to 10 °N) of (a) zonal wind stress induced Ekman pumping/suction in JRA55-do forcing and (b) the associated bias, (c) zonal wind stress induced Ekman pumping/suction in CORE forcing and (d) the associated bias, (e) wind stress curl induced Ekman pumping/suction in JRA55-do forcing and (f) the associated bias, and (g) wind stress curl induced Ekman pumping/suction in CORE forcing and (h) the associated bias.





in zonal wind near the ITCZ creates a positive wind stress curl with Ekman suction that also compensates the effect of the
      easterly wind on generating the Ekman pumping. Due to the compensating effect, the bias of zonal wind at around 120°W in
      the CORE simulation is compensated by the bias in wind stress curl [figure 17d,h]. In other words, the Ekman mechanism at
      120°W in the CORE simulation, though close to the observations like JRA55-do, is right for the wrong reason. Particularly
      in the tropical Pacific, this compensation effect should be further examined in the future when evaluating simulations to better
understand the underlying biases.

## 5.2   Surface heat fluxes

We now examine the thermodynamical contribution from the surface heat flux to the seasonal DSL variation in JRA55-do. The
latent heat flux and solar radiation control the net surface heat flux at the seasonal time scale [figure 18a-c]. At the seasonal
time scale sea surface temperature (SST) is determined by the surface net heat flux with a roughly three months time lag [figure
18a,d]. The mixed layer depth follows the SST which deepens during low SST because of the reduced stratification near the
ocean surface and shoals during high SST because of the enhanced stratification [figure 18d,e]. This mixed layer pattern can
not be used to explain the spatio-temporal pattern seen in DSL seasonal variation [figure 16]. Instead, we find that the 20°C
isotherm shows the same wave propagation as the DSL, thus indicating the baroclinic nature of this seasonal Rossby wave
propagation and the dominant role of ocean dynamics on the DSL variation. The seasonal analysis over the tropical Pacific
shows the important role of ocean dynamics on the DSL variation which is quite different from the mid-latitude where the
thermodynamic forcing dominates the changes of seasonal DSL (Vinogradov et al., 2008).

## 5.3   Zonal currents

Due to the significant seasonal fluctuations in DSL within the tropical Pacific, we use the same box budget as in Figure 5 to
study the corresponding zonal current variations between Wbox and Ebox. The zonal current anomaly is strongly affected by
the meridional DSL gradient. In the 2°N to 10 °N zonal band, we find one of the largest DSL variance. The corresponding
meridional DSL gradient changes, in response to the Rossby wave propagation, can have a large impact on the seasonal changes
of the NECC. Figure 19a,b shows the seasonal amplitude and phase defined by the largest values of the monthly climatology
and the corresponding month, respectively. The SEC and the EUC show comparable and dominating roles on seasonal heat
advection between Ebox and Wbox. The seasonal amplitude of heat advection in the CORE simulation is larger than with
JRA55-do for all currents, which is consistent with larger wind stress forcing and DSL amplitude in CORE.

All seasonal phases agreed between the two simulations except for the NECC and SEC. The SEC difference between the
two simulations is mainly due to the small difference in the sub-seasonal signal that causes the difference in phase [figure 19c].
For the NECC, the difference in phase can also be seen in the volume transport, which peaks in June for CORE and November
for JRA55-do [figure 19b,c]. The simulation forced by JRA55-do shows a better agreement with the observed NECC which
peaks in December (Johnson et al., 2002).

The zonal current analysis confirms the importance of an accurate simulation of the DSL at the seasonal time scale. This
analysis also demonstrates the crucial role of the surface wind stress timing near the eastern tropical Pacific, which influences





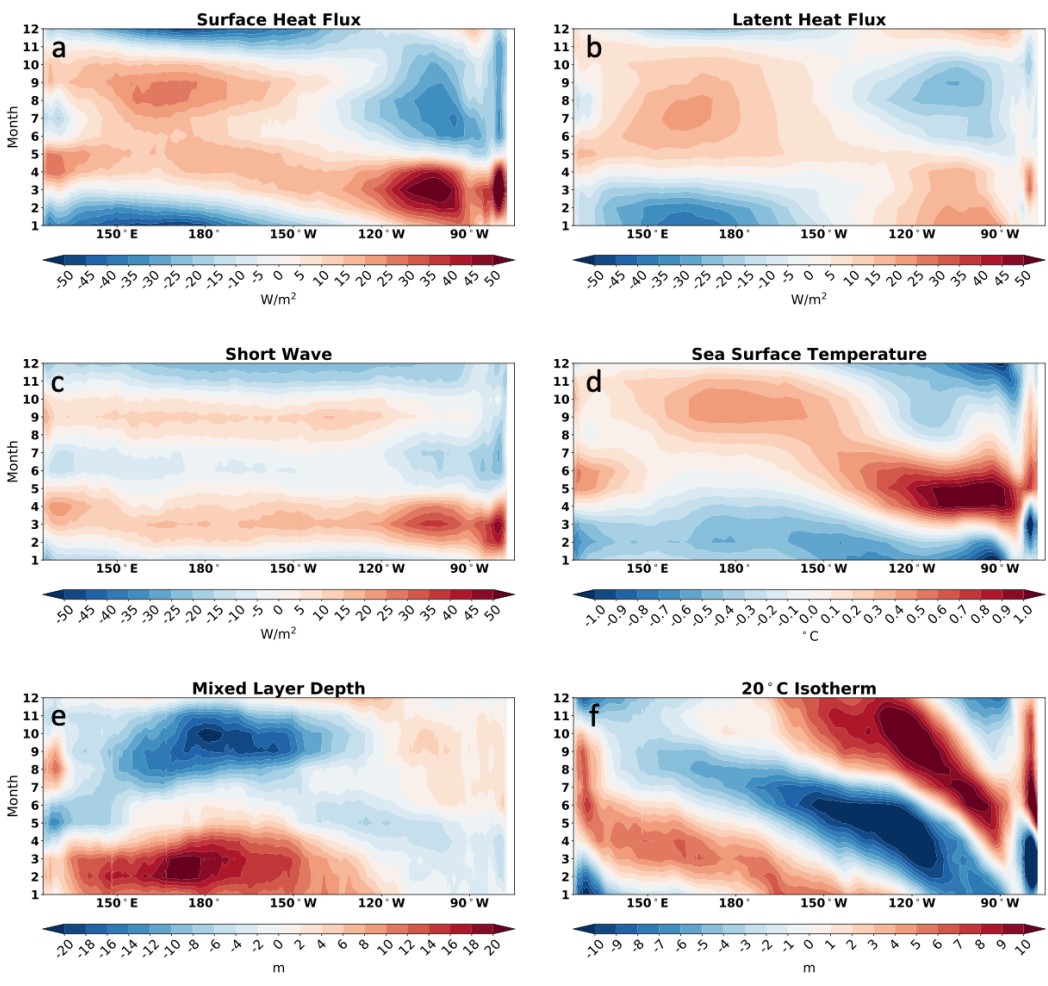

**Figure 18.** Hovmöller diagram of monthly climatology (1993-2007) with mean state removed showing the meridional mean (2°N to 10 °N) of (a) surface net heat flux, (b) latent heat flux, (c) short wave radiative flux, (d) sea surface temperature, (e) mixed layer depth, and (f) 20°C isotherm from the JRA55-do forced simulation.



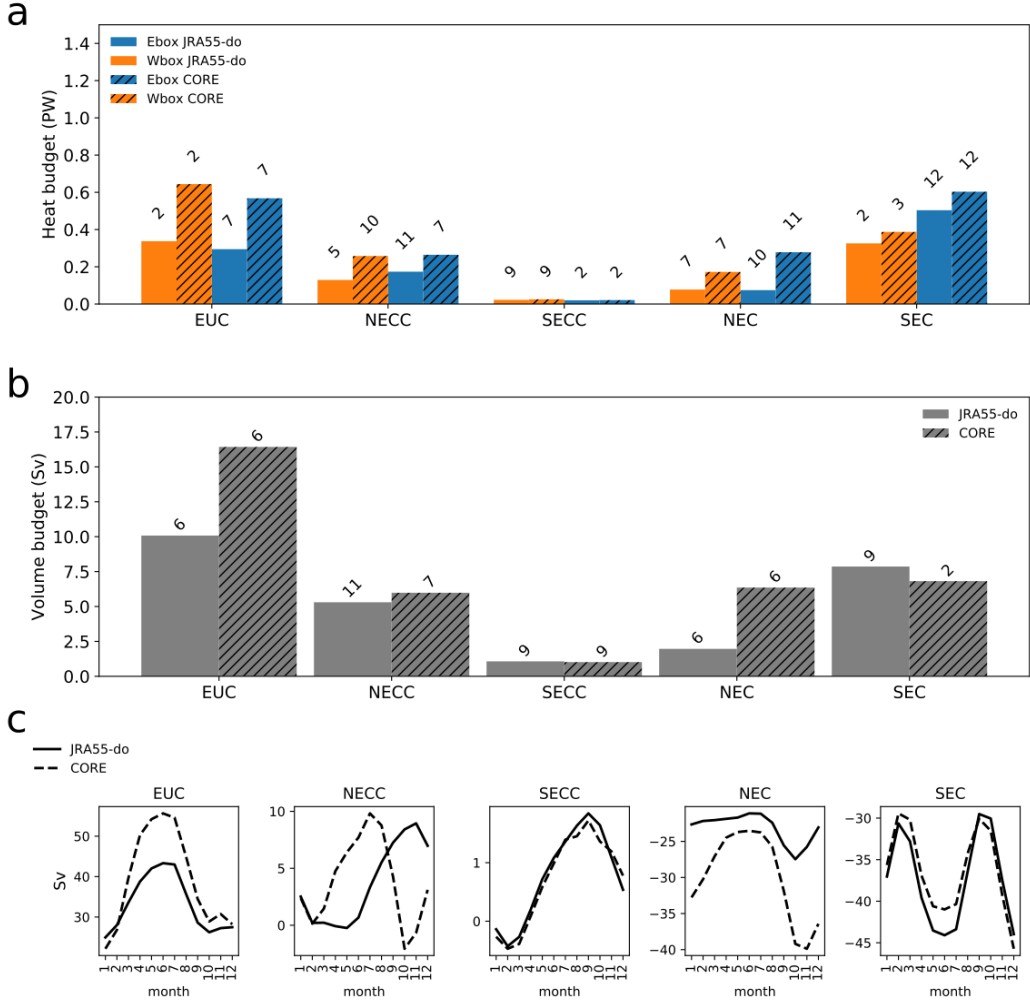

**Figure 19.** The seasonal amplitude (bar) and phase (integer which represents month on top of each bar) of (a) heat advection to the Wbox (yellow bar) and Ebox (blue bar) across the 180° transect and (b) the volume transport (gray bar) from individual current. The JRA55-do forced simulation (no stripe) and the CORE forced simulation (striped) are place side by side for comparison. (c) the seasonal variation of volume transport from JRA55-do (solid) and CORE (dashed) for each current.





the timing of heat advection and volume transport across the tropical Pacific. The NECC and SECC, though smaller than SEC and EUC in volume transports, are the only two currents that show a change of direction in the simulation at seasonal time
scales [figure 19c]. This current reversal, however, does not exist in the observations. In the simulations, the current reversals results from the underestimated mean state of the currents that leads to the incorrect seasonal heat and mass transport direction. However, due to the small contribution in volume and heat budgets from these two currents, the model simulation is not greatly affected by this reversal at the seasonal time scale.

## 6 Dynamic Sea Level Variability during El Niño

Besides the seasonal variation, one of the largest DSL fluctuations over the tropical Pacific is related to the El Niño-Southern Oscillation (ENSO) at the interannual time scale. To investigate the DSL bias during El Niño, we first define and find all El Niño events in both simulations. We use the Oceanic Niño index (ONI) to find all El Niño events during 1958-2007. To obtain the ONI, we calculate the area-weighted mean of monthly SST anomalies in the Niño3.4 region (5°N - 5°S, 170°W - 120°W) with seasonal climatology and long-term trend removed before calculating the 3-month running mean. The detrending follows
the method from the Climate Prediction Center at NOAA by removing the 30-year means from every five years centered at the 30-year window. The El Niño event is defined when ONI has 5 consecutive values that are larger than 0.5°C and ended when the ONI is lower than 0.5°C. Both simulations show good agreement with the observed ONI [figure 20a]. To better describe El Niño stages, we define Year0, Year1, and Year2 based on the composite El Niño period [figure 20b] where Year1 winter is when ONI reach maximum, Year0 is one year before Year1, and Year2 is the following year of Year1. A composite of DSL
and wind stress from simulations and observations are calculated based on a total of 12 El Niño events during the 1958-2007 period [figure 20b]. To determine the composite, we remove the mean, trend, and seasonal signals in all of the time series.

### 6.1 Oscillator theories

We first focus on the zonal mean of the DSL variation in the tropical Pacific region during El Niño. A clear recharge-discharge oscillator affecting DSL can be seen evolving during El Niño [figure 21a,c,e] (Jin, 1997). According to the oscillator theory,
the warm water volume continues to be charged into the equatorial region before it reaches the peak of an El Niño. It then discharges to higher latitude after the peak as a result of the Sverdrup transport response to wind forcing over the tropical Pacific. The SST lags the DSL positive anomaly before the peak and the negative anomaly after the peak of El Niño. This lag means that the subsurface processes build-up the warm water volume before the SST starts to warm up during the charging stage. During the discharging stage, the subsurface processes release the warm water volume before the SST starts to cool
down.

Consistent with the theory, DSL reaches the peak at the end of Year 1 for both simulations during the charging stage which increase the warm water volume near the equator and quickly changes to a negative anomaly while the higher latitude changes from a negative to positive anomaly, which indicates the start of the discharging stage [figure 21a,c]. The lag of the SST is also clear in both simulations. The CORE simulation shows a larger DSL bias which is consistent with the bias shown at other time







**Figure 20.** (a) The Ocean Niño Index (ONI) in JRA55-do (blue solid), CORE (orange solid), and observation provided by the Climate Prediction Center at the National Oceanic and Atmospheric Administration (black dashed). (b) All El Niño events picked out during the period of 1958-2007 to calculate the El Niño composite.



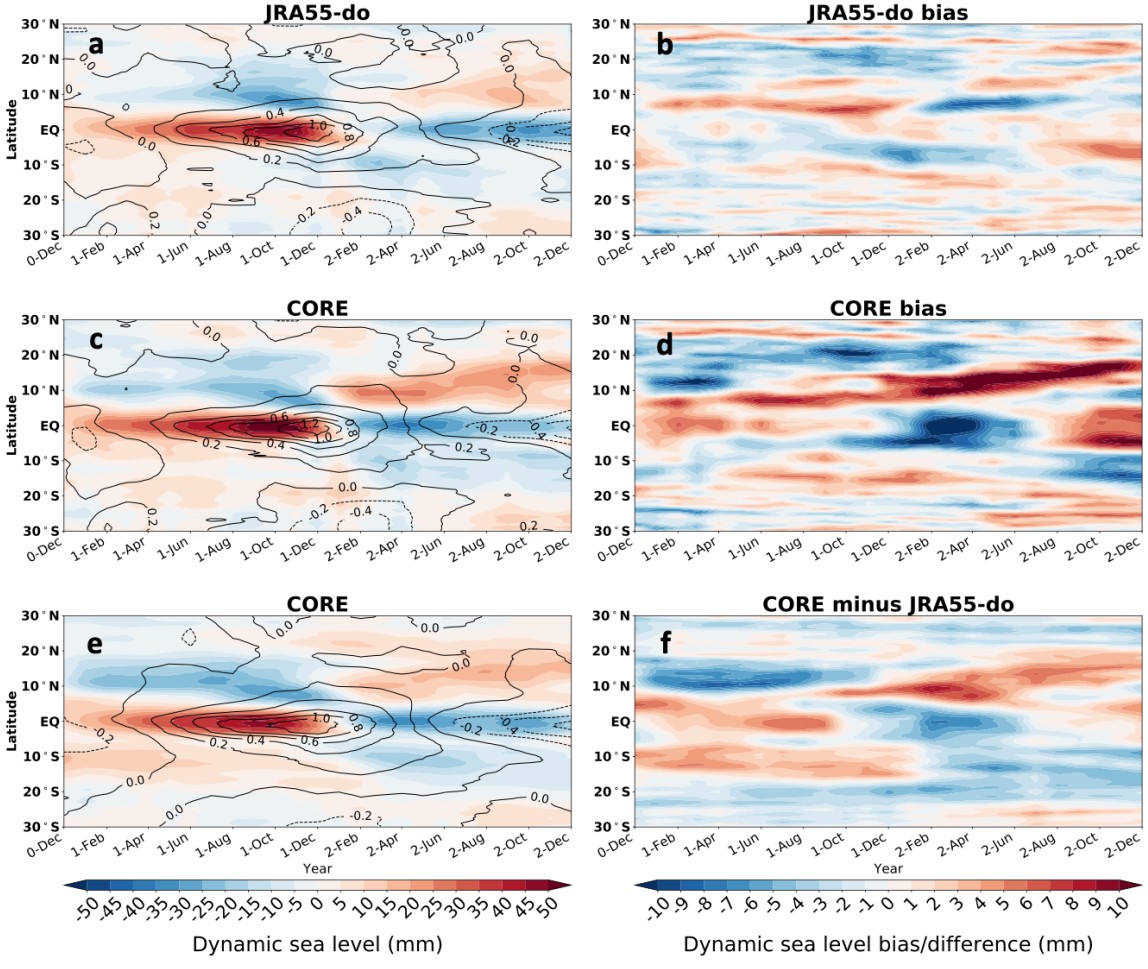

**Figure 21.** Hovmöller diagram showing the zonal mean over the tropical Pacific basin during the El Niño composite (1993-2007 with a total of four El Niño events) in the (a) JRA55-do simulated DSL variation (shading), sea surface temperature (contour with unit °C), and (b) the associated DSL bias and (c) CORE simulated DSL variation (shading), sea surface temperature (contour with unit °C), and (d) the associated DSL bias. (e) Hovmöller diagram of the zonal mean over the tropical Pacific during the El Niño composite (1958-2007 with a total of 12 El Niño events) in the CORE simulated DSL variation (shading), sea surface temperature (contour with unit °C), and (f) the JRA55-do simulated DSL subtracted from CORE.



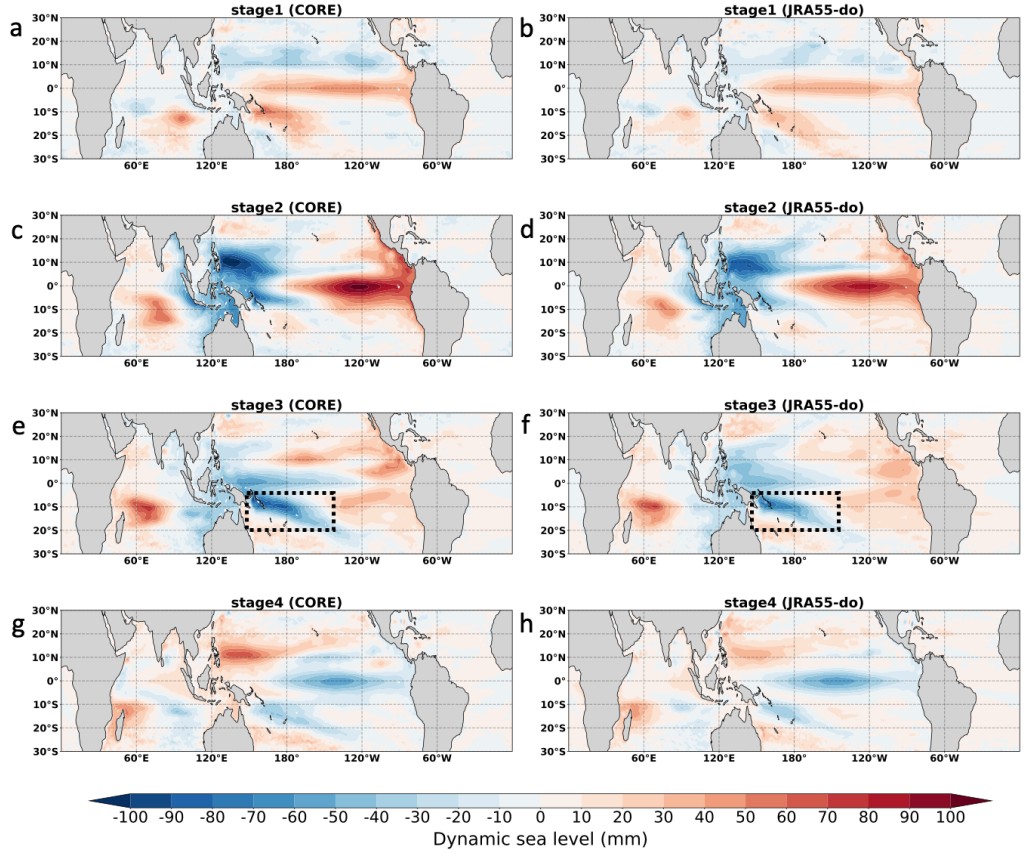

**Figure 22.** The DSL anomaly during the El Niño composite as determined during 1958-2007 in four different stages. The first row shows the first stage (the mean of 7-12 months prior to the maximum ONI value) in (a) CORE and (b) JRA55-do. The second row shows the second stage (the mean of 6 months prior to the maximum ONI value) in (c) CORE and (d) JRA55-do. The third row shows the third stage (the mean of 6 months following the maximum ONI value) in (e) CORE and (f) JRA55-do. The black dashed boxes show the region of asymmetric sea level changes in the southwestern tropical Pacific. The last row shows the fourth stage (the mean of 7-12 months following the maximum ONI value) in (g) CORE and (h) JRA55-do.

scales [figure 21b,d]. The recharge-discharge pattern can also be seen during the longer period composite (1958-2007) [figure 21e].

We separate the El Niño period into four different stages [figure 22]. The first is the initiation stage calculated as the mean of seven months to twelve months prior to the maximum ONI (February, Year1 to July, Year1). In this stage we see the positive DSL anomaly initiated in the central tropical Pacific along the equator. The second stage is the mean of six months prior

to the maximum ONI value (August, Year1 to January, Year2). The positive DSL anomaly propagates eastward due to the propagation of an equatorial downwelling Kelvin wave while the negative DSL anomaly strengthens in the western tropical Pacific in both hemispheres (Zebiak and Cane, 1987). Based on the ENSO oscillator theories, the negative DSL anomaly in





the western tropical Pacific is a result of two phenomena. One is the upwelling Rossby wave propagating westward on two sides of the equator reaching the western boundary based on the delayed oscillator theory (Zebiak and Cane, 1987). The other
is related to the shoaling of the off-equator thermocline based on the western tropical Pacific oscillator theory (Weisberg and Wang, 1997). The latter also leads to the increase of sea level pressure due to decreasing SST and the associated easterly wind at the equator in the third stage.

The third stage is the mean of six months following the maximum ONI value (January, Year2 to June, Year2). The positive anomaly in the eastern Pacific starts to dissipate through a coastal Kelvin wave moving heat toward the poles (Johnson and
O'Brien, 1990). The reflected downwelling Rossby wave on two sides of the equator, on the other hand, tries to move the heat back to the warm pool in the western Pacific (Picaut et al., 1997). Like in stage 2, the DSL drop in stage 3 over the western tropical Pacific is also a result of two phenomena. Based on the delay oscillator theory, the reflected upwelling Rossby wave related DSL drop in the western tropical Pacific starts showing at the equator. On the other hand, the western tropical Pacific oscillator theory emphasizes the importance of equatorial easterly winds, due to the positive sea level pressure off-equator,
on generating the negative DSL anomaly at the equator. This stage is also referring to the discharging stage in the recharge-discharge oscillator theory where the warm water volume is discharged poleward from the equator. The final stage is calculated as the mean of seven months to twelve months following the maximum ONI value (July, Year2 to December, Year2). In the final stage, the downwelling Rossby wave on two-sides of the equator shows in the western tropical Pacific for both simulations.

In general, the model simulations show great resemblance in terms of spatial patterns and temporal changes. Both simulations
demonstrate the oscillator theories well. However, asymmetry in the DSL changes on two sides of the equator seem missing from the theories mentioned above during El Niño. The asymmetric pattern, like the negative DSL signal in the southwestern tropical Pacific, is related to other mechanisms dominating the DSL signal over certain regions, which reduces the symmetry along the equator.

## 6.2 Asymmetry in the dynamic sea level along equator

The negative DSL anomaly and the asymmetry of DSL in the southwestern tropical Pacific are not due to the lack of discharging of warm water from the equator, but instead due to the drop in DSL signal that is intensified after the peak ONI (stage 3) [figure 22e,f]. This dropping DSL signal can also be found in the composite based on observations during 1993-2007 [figure 23]. We see that the Southern Hemisphere DSL drop is strongly related to the pressure difference between Australia and the Southern Hemisphere central to eastern tropical Pacific [figure 24]. This pressure difference, known as the Southern Oscillation Index
(SOI), causes a negative wind stress curl in the region of DSL drop in stage 2 [figure 24c,d] and stage 3 [figure 24e,f], which results in Ekman suction and the associated negative DSL anomaly.

The negative DSL anomaly in the southwest tropical Pacific started before the ONI peak (stage 2) and reaches maximum right after the peak (stage 3) [figure 22]. In stage 2, a high-pressure center is established near Australia while a low-pressure center is formed near the southeast tropical Pacific [figure 24c,d]. Between the two pressure systems, a strong geostrophic wind
toward the equator is formed and turns to the right near the equator due to the low pressure at the eastern equatorial Pacific. This



**Figure 23.** The DSL evolution from satellite altimeter observation in four different stages (top to bottom) during the El Niño composite (1993-2007) defined as figure [22]. The black dashed box shows the region of asymmetric sea level change in the southwestern tropical Pacific at stage 3.





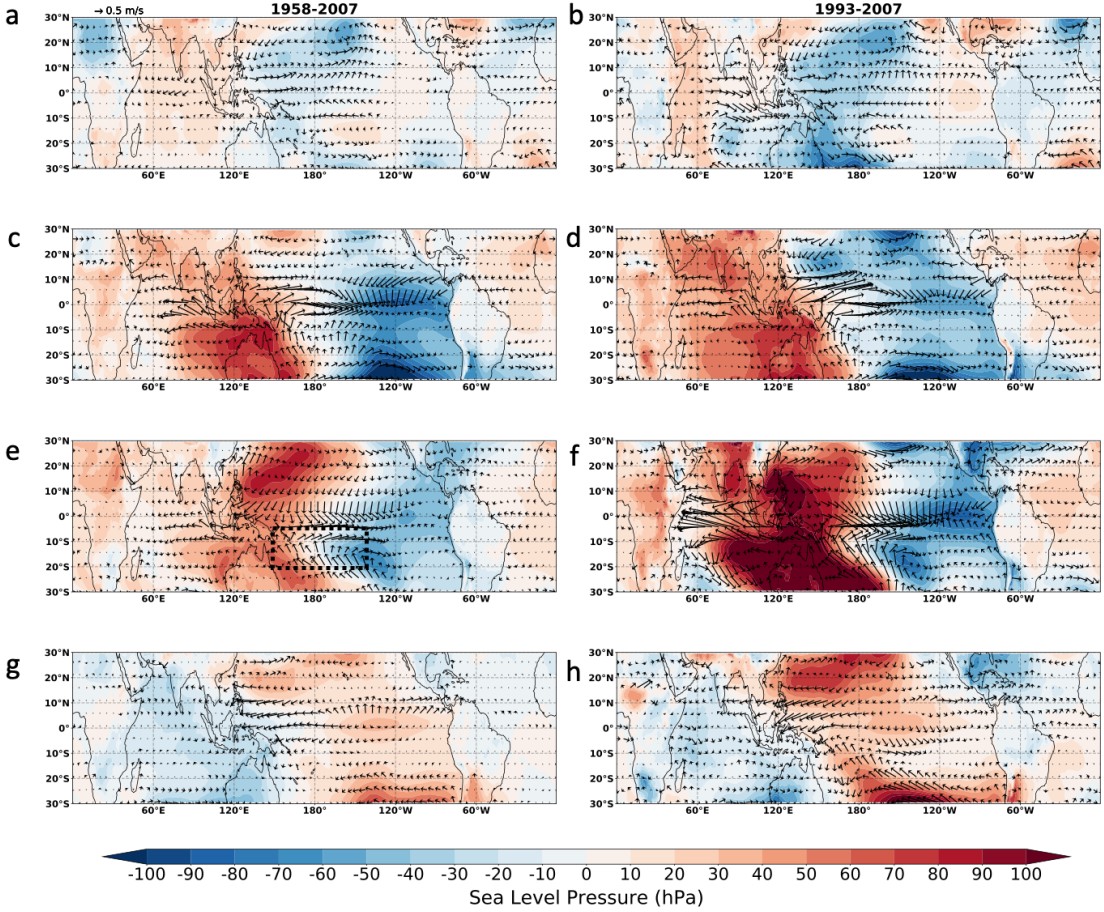

**Figure 24.** The sea level pressure (shading) and surface wind (vector) from JRA55-do forcing in four different stages (top to bottom), defined as figure [22], during the El Niño composite determined over (a,c,e,g) 1958-2007 and (b,d,f,h) 1993-2007. The black dashed box shows the region of asymmetric sea level change in the southwestern tropical Pacific at stage 3.

turning creates a negative wind stress curl in the southwestern tropical Pacific that causes Ekman suction and the associated negative DSL anomaly.

After the ONI peak, a low-pressure center is established in the south-central tropical Pacific [figure 24e,f]. This low-pressure center, closer to the southwest tropical Pacific than in stage 2, causes a stronger negative wind stress curl than in stage 2 due
to the increased pressure gradient that results in stronger Ekman suction and DSL drop. Eventually, the dropping DSL signal subsides due to decreased and reversed pressure contrast related to the Southern Oscillation [figure 24g,h]. Since the composite based on observations only includes four El Niño events, the different sea level pressure amplitudes could be readily dominated by the strongest events. Therefore, the amplitude changes between longer (1958-2007) and shorter periods (1993-2007) should not be seen as a long term trend but instead a sign of possible decadal variability of the ENSO/SOI strength.



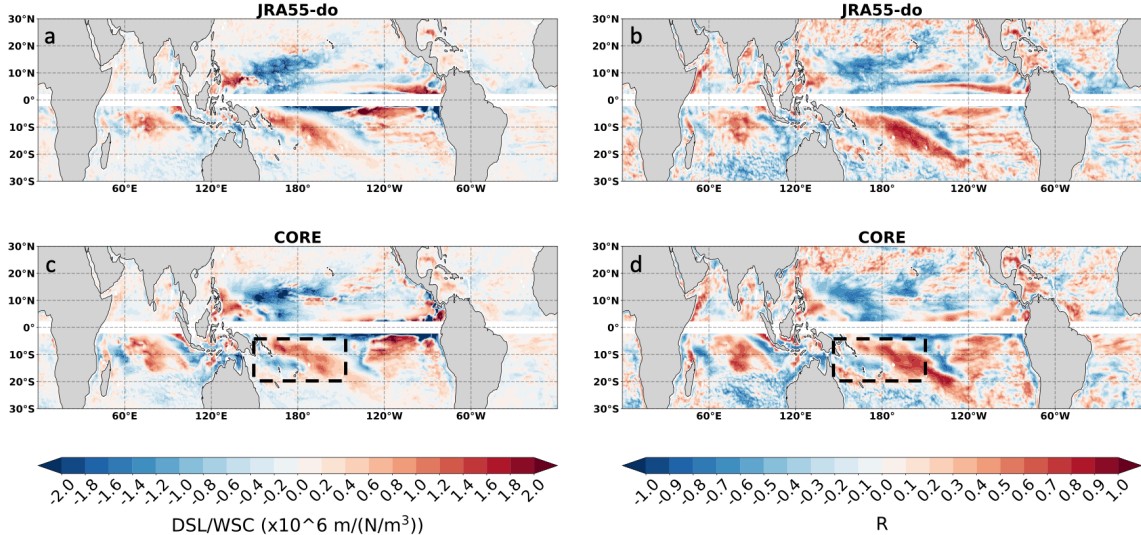

**Figure 25.** The regression coefficients between wind stress curl and DSL anomaly during the El Niño composite (1958-2007) in (a) JRA55-do and (c) CORE. The R value from the regression in (b) JRA55-do and (d) CORE. The black dashed boxes show the region of asymmetric sea level changes in the southwestern tropical Pacific.

The mechanism of the asymmetric sea level along equator is missing in the known oscillator theories. The delayed oscillator theory can explain the DSL drop in the western tropical Pacific resulting from a westward propagation of upwelling Rossby wave in stage 2. The western tropical Pacific oscillator, on the other hand, emphasizes the importance of dropping SST in the western tropical Pacific due to the shoaling of the thermocline (dropping DSL) as a result of cyclonic wind generated from off-equatorial convection in stage 2. (Deser and Wallace, 1990; Weisberg and Wang, 1997). Both theories cannot explain the

maximum value of DSL drop in stage 3 in the Southern Hemisphere and the asymmetric response of DSL and sea level pressure between the north and south.

To better quantify the role of the wind stress curl on the DSL variation in the tropical Pacific during the composite period, we regress the wind stress curl with the DSL anomaly at each grid point [figure 25]. Figures 25b,d show the strong positive correlation (R > 0.6) between DSL and wind stress curl in the region where the negative DSL anomaly is located [figure 22]

in both JRA55-do and CORE simulations. The regression shows a regression coefficient of around $1.2 \times 10^{-8}$ N m$^{-3}$ per 10-millimeter DSL change. Comparing with the Southern Hemisphere, the maximum DSL drop in the Northern Hemisphere happens in stage 2. This drop is quickly restored toward the mean state in stage 3 [figure 22] due to an increase of negative wind stress curl created by the high sea level pressure [figure 24], which is a process described by the western tropical Pacific oscillator theory (Weisberg and Wang, 1997). This behavior is also shown in the regression maps where the wind stress curl is

negatively correlated with the DSL anomaly in the region from 150°E to 180° and equator to 20°N.





This analysis demonstrates the importance of sea level pressure evolution in the tropical Pacific on the DSL changes during El Niño. In stage 2, the combination of ocean wave propagation (delayed oscillator) and wind stress curl creates the maximum DSL drop in the northwest tropical Pacific. In stage 3, the air-sea interaction (western tropical Pacific oscillator) quickly restores the northwest tropical Pacific DSL to the mean state through the established high sea level pressure, while the negative wind
stress curl established by the sea level pressure gradient in the Southern Hemisphere creates the maximum DSL drop in the southwest tropical Pacific. The different mechanisms between north and south indicate the lag in DSL signal is due to different forcings. It is not an energy redistribution or wave propagation between northwestern and southwestern tropical Pacific. This asymmetry in the DSL dropping signal between the north and south of the western tropical Pacific can usually be seen during El Niño.

**7 Summary and Conclusions**

In this study, we use the GFDL-OM4 global ocean-sea ice model, as forced by JRA55-do and CORE, to investigate the DSL variability and the associated biases when comparing to available observational datasets. We are able to show the improvement found with the JRA55-do forcing used in OMIP-II relative to the CORE forcing in OMIP-I. We also reveal needed future improvements in the forcing across different time scales.

**7.1 Summary**

The mean state bias of DSL persists in the simulations forced by JRA55-do when comparing to CORE. The missing DSL trough along 9°N in both CORE and JRA55-do forced simulations shows the importance of sharp zonal wind stress changes near the ITCZ. In the 4°N to 9°N latitude band, the bias in the wind stress forcing causes biases in the geostrophic current that leads to the flattening of the DSL gradient in the meridional direction. The bias in the geostrophic current directly impacts the
mean strength of NECC and NEC. A future improvement of the zonal wind shear in the tropical Pacific is needed for a better DSL and zonal current representation in ocean model simulations.

The JRA55-do forced simulation significantly improves the DSL trend bias over the tropical Pacific. The improved zonal wind stress trend in JRA55-do over the eastern equatorial Pacific is the main reason for the better DSL trend simulations. The trend analysis shows extra attention is needed on the method used for correcting the magnitude of the wind stresses found in
reanalysis data. A single multiplicative factor applied to the entire time series of wind stress may not be appropriate since it can cause errors in variability and changes in other time scales. A DSL trend bias along 10°N, though improved, still exists in the JRA55-do forced simulation. We identify an easterly wind trend bias, related to this persisting DSL trend bias in both CORE and JRA55-do, as a point of focus for improvements in the atmospheric forcing.

The JRA55-do forced simulation also shows improved seasonal DSL variation due to a better representation of the wind
stress curl forcing in the eastern tropical Pacific. The improved timing of wind stress curl in JRA55-do results in a more accurate Rossby wave propagation in the 0-10°N zonal band, thus leading to an improved NECC simulation. In this zonal band, both zonal wind stress and wind stress curl contribute to the Ekman pumping/suction with compensating effects at the





seasonal time scale. Though the final effect of Ekman pumping/suction in CORE is similar to JRA55-do, the CORE forcing data has a relatively inaccurate representation of the zonal wind and wind stress curl. The reason for the similar Ekman
pumping/suction is mainly due to the compensation effect in CORE between the bias in the zonal wind and the bias in wind stress curl. Detailed tropical ocean analysis is needed on the effect of Ekman pumping/suction in future model evaluation to avoid ignoring this bias compensation.

Both CORE and JRA55-do generate reasonable DSL variations during El Niño compared to observations, with smaller biases in the JRA55-do simulation. An asymmetry in the DSL change during an El Niño event on two sides of the equator has
not been explained in the existing oscillator theories. We find the asymmetry in the DSL pattern is strongly related to wind stress curl that follows the sea level pressure evolution during El Niño. An atmospheric sea level pressure gradient created by a pressure drop in the southeastern tropical Pacific and a pressure rise in Australia as well as the sea level pressure drop in the cold tongue region are the main contributors to the negative wind stress curl that drives the DSL drop. The high correlation between wind stress curl and DSL shows the dominant influence of local wind forcing on the DSL signal. The analysis demonstrates
the importance of the external forcing on the off-equator DSL changes during an El Niño event.

### 7.2   Recommendations for key bias reductions

To reduce the time mean bias of DSL in future OMIP simulations over the tropical Pacific, accurate zonal wind stress shear near the ITCZ is crucial. Especially over the 4°N to 9°N zonal band, the biases in the time mean of the DSL can further affect the mean ocean current strength and cause an artificial seasonal ocean current reversal.

For the bias in the long-term trend of DSL, the JRA55-do has significantly improved the westerly wind bias at the eastern Pacific near the equatorial region (10°S to 10°N) that results in the reduction of DSL bias along 10°N. To further reduce this DSL bias along 10°N, we find easterly trend biases exist along 20°N in both the JRA55-do and CORE forcings that create the artificial positive wind stress curl along 10°N, resulting in the negative sea level bias. The multiplicative factor applied to correct the wind stress is highly related to the wind stress bias. A careful evaluation of the wind stress field across different
time scales after applying the factor is necessary to reduce the bias of the simulated DSL.

For the seasonal variation of the DSL, the dominant external forcing is region-specific. Over the deep tropics, from the equator to around 10°N/S, ocean dynamics are the dominating factor which was mainly initiated at the eastern boundary near 90°W due to Ekman suction/pumping induced by local wind stress curl. Therefore, capturing the timing and strength of the local wind stress curl is crucial to better simulate the seasonal variation of the DSL. The accurate DSL variation can further
improve the simulated ocean current in seasonal time scale, like the improved NECC in the JRA55-do comparing to the CORE forced simulation. However, the continuously underestimated DSL amplitude in the JRA55-do simulation, which is related to the underestimated wind stress curl in the same zonal band needs further improvement. Besides the local boundary wind forcing near 90°W, the underestimated seasonal zonal wind stress variability between 150°W and 180° in both JRA55-do and CORE causes the underestimated DSL signal throughout the year. Due to the compensation effect of zonal wind stress and
wind stress curl on the strength of Ekman suction/pumping in the tropics, a separated evaluation of both zonal wind stress and wind stress curl is needed in the future to avoid the compensated bias shown in the CORE forcing data.



For interannual variability, the El Niño related DSL variation is well represented in JRA55-do and CORE for both the timing and the spatial pattern. However, we still see a amplitude bias for DSL in both CORE and JRA55-do forced simulations. The JRA55-do does improve the amplitude bias when comparing to the CORE.

The multiple time scale analyses in this study provide a comprehensive view of important variability, changes, and the associated biases in the tropical Pacific. Any improvement of the biases mentioned in this study will be very helpful to further a mechanistic understanding of tropical Pacific DSL patterns using OMIP simulations, and by extension their related coupled climate models.

*Data availability.* The forcing dataset, JRA55-do, is available through input4MIPs (https://esgf-node.llnl.gov/search/input4mips/, last ac-
cess: 26 May 2020). The observed sea surface height is from Copernicus Marine Environment Monitoring Service (CMEMS) (https://resources.marine.copernicus.eu/, last access: 1 July 2019). The EN4 dataset is from the Met Office Hadley Centre (https://www.metoffice.gov.uk/hadobs/en4/, last access: 13 August 2019). The Wave and Anemometer-based Sea Surface Wind is downloaded from the Kyushu University (https://www.riam.kyushu-u.ac.jp/oed/tokinaga/waswind.html, last access: 19 August 2019). An archive for all of the model outputs is available through CMIP6 project under OMIP. (https://esgf-node.llnl.gov/search/cmip6/)

*Author contributions.* CWH, JY, SMG proposed and led this evaluation study. CWH performed the analyses including processing the model outputs and observational data. RD produced the model outputs. All authors contributed to the writing and editing processes.

*Competing interests.* The authors declare that they have no conflict of interest.

*Acknowledgements.* This study is supported by the NOAA Climate Program Office (grant NA18OAR4310267). We thank Rebecca Beadling and Andrew Wittenberg for very useful comments on an earlier draft of this manuscript. We acknowledge the World Climate Research
Programme coordinating CMIP6 and the related activities, like OMIP and input4mips. We thank the archiving effort done by Earth System Grid Federation. We thank the EU Copernicus Marine Service Information providing the Sea surface height data from multiple satellite altimetry measurements. We thank the Met Office Hadley Centre archiving the temperature and salinity data. We thank Dr. Hiroki Tokinaga on processing and archiving the WASwind data.



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
