# Peer review of "A Mechanistic Analysis of Tropical Pacific Dynamic Sea Level in GFDL-OM4 under OMIP-I and OMIP-II Forcings"

_Geoscientific Model Development, 2020_

## Referee Comment (RC1) · Anonymous Referee #1 · 19 Jan 2021

General comments

In this paper, authors present dynamical features of tropical Pacific sea level simulated by GFDL-OM4 under OMIP protocols. Authors investigate biases of simulated dynamic sea level (DSL) forced by the distinct surface atmospheric datasets prepared for two phases of OMIP (CORE and JRA55-do for OMIP-1 and OMIP-2, respectively). They take up the following aspects of tropical Pacific DSL field that characterize its mean state and variability.

- Time mean (Section 3).

- Decadal and longer trend (Section 4).

- Seasonal variability (Section 5).

- Response to interannual ENSO variability (Section 6).

Above aspects of the tropical Pacific DSL is largely determined by imposed wind stress forcing. The long-term mean DSL has positive bias around the intertropical convergence zone in both OMIP-1 and OMIP-2 simulations. This is caused by the biases in wind stress forcing which have been introduced into the forcing dataset by the method to adjust wind vectors. The long-term trend over the northern tropical Pacific is improved in OMIP-2 relative to OMIP-1, but it is still suffered from a negative bias due to easterly wind trend along 20N. Seasonal variability of the North Equatorial Counter Current, which is caused by the seasonal variability of DSL, is also improved in OMIP-2. Both OMIP-1 (CORE) and OMIP-2 (JRA55-do) forcing datasets generate realistic DSL variation during El Niño/Southern Oscillation (ENSO), which includes a meridional asymmetry across the equator.

I think that results are presented well overall and I do not find any serious issues in analysis and reasoning. However, I am also a bit afraid that the paper may give some readers an impression that the presented contents are somewhat superficial. I thought that in several places descriptions are given without providing robust evidences and thus not easy to follow. Specifically, the discussion about the cause of the absence of the observed DSL trough around 9N in simulations is made using zonally averaged zonal wind (e.g., Fig.4b). Considering the slanting distribution of the wind convergence zone in the eastern tropical north Pacific, two-dimensional distribution of wind stress curl anomaly might be more illustrative. The seasonal variation of NECC might be more clearly explained by using seasonal evolution of horizontal distribution of DSL in the tropical north Pacific. Also, I thought it would be helpful for the reader if authors add some paragraphs that give quantitative guidance about how the simulated biases may compromise practical assessments of sea level variability based on the OMIP simulations. For example, are the biases of DSL trend comparable to global mean sea level rise observed in recent years? I would like to ask the authors to add or

revise materials used to explain important features. This would attract a wider range of readers.

Specific comments

L129: "DSL correlates well with the surface wind stress in the mean state". I think that this expression is somewhat inappropriate. I would suggest something like, "distribution of DSL can be explained well by the surface wind stress in the mean state", if I understand the authors' intention correctly.

Figures 3 and 4: Given that the locations of wind stress convergence are slanting from southwest to northeast in the eastern north Pacific as shown in Figure 3a, contribution of zonal derivative of the meridional component to wind stress curl might not be neglected and the horizontal distribution of wind stress curl bias would be of interest. Is the bias pattern of either wind stress curl or Sverdrup stream function comparable with the DSL bias shown in Figures 3b and 3c? A bit more detail would clarify the point that should be improved in the forcing dataset.

L.213-215: "We hypothesize that the key reason for the weak NECC is due to both the underestimated zonal wind stress in JRA55-do and a flattening of the DSL trough due to the wind stress curl bias in the northern tropical Pacific found in both CORE and JRA55-do [figure 4b,c]." I think that a bit more detailed explanation is required about how the underestimated zonal wind stress in JRA55-do is related to the weak NECC.

L.251: Does the excessive westerly wind trend in CORE and JRA55-do affect simulated features of the global warming hiatus?

L.374-375: How does the improved seasonal variation of DSL in JRA55-do result in the better seasonal variation of NECC?

L.493-495: "the bias in the wind stress forcing causes biases in the geostrophic current that leads to the flattening of the DSL gradient in the meridional direction". I think that the flattening of the DSL leads to the biases in the geostrophic current. For example, "the bias in the wind stress forcing causes flattening of the DSL gradient in the meridional direction that leads to biases in the geostrophic current".

Technical comments

L58-59: It would be worth noting here that baroclinic deformation radius in the tropics can be resolved by the 0.25-degree resolution used by GFDL-OM4 as also noted in L.122-123.

L180-181: "the heat is not stored in the eastern tropical Pacific but flushed to the western tropical Pacific". I think this is worth noting here that "details are discussed in the next paragraph".

L.218-220: I think discussion about trend can be moved to somewhere around the paragraphs that discuss DSL trend using Figure 8 in section 4.

L.252: "extends" should read "extending".

L.266: "(called offsetting factor)". That is actually used as an offsetting factor to correct JRA-55 wind.

L.314: "off" should perhaps read "offset" or "delayed".

---

## Referee Comment (RC2) · Anonymous Referee #2 · 4 Mar 2021

Review on manuscript (#gmd-2020-374) "A Mechanistic analysis of tropical Pacific dynamic sea level in GFDL-OM4 under OMIP-I and OMIP-II forcings" by Hsu, Yin, Griffies and Dussin

This manuscript examined sea level simulation in the tropical Pacific by the GFDL-OM4 ocean model driven by two different atmospheric products: CORE and JRA55-do. Long-term mean, linear trends, seasonal and interannual variability are discussed in detail, in particular focusing on the differences between model results and observations, and between two model experiments. It's found that the JRA55-do tends to give improved simulation, closer to observations than CORE.

Generally speaking, the manuscript was written fairly well and easy to follow. Detailed comments are as follows.

General Comments

1. My major comment/concern after reading through this quite lengthy manuscript is that I feel it may be more appropriate for other journals than GMD – a journal focusing on model development. But I have to say it's my personal feeling based on my understanding of GMD versus other journals, and it's really up to the editor to decide.

If the authors agree on this, I suggest the authors can reduce some parts (which don't really show new results), and tighten up the storyline. I guess it would be a better paper.

2. Wind stress and wind stress curl are heavily discussed throughout this paper, which is fine. However, the coarse-resolution (4x4 deg) WASwind product is used, and differences between JRA55-do and CORE are defined as "biases". I feel it could be problematic. For example, some features in the wind stress curl map (Fig. 11b) may be "artificial", reflecting the 4x4 grid. I think there should be some better-quality wind products, e.g., based on satellite Scatterometer observations?

detailed comments

Line 23 (L23), this sentence is confusing and thus needs to be changed. Sea level variability is not only associated with ocean temperature (heat content), but can also due to halosteric component (ocean salinity) and mass component.

L31, OMIP-I and II are not defined before. I feel they should be introduced here with a couple of sentences. What are main differences and similarities between them? It may help to state your motivation more clearly. Some material from the 2nd paragraph of Section 2.1 (L63-69) can be moved here.

L96-97, steric sea level rise is not only due to ocean warming, but also ocean freshening (decreasing salinity).

L105, change "which relates" to "which mainly relates", since salinity can also play a role, in addition to ocean temperature (or equivalently thermocline depth change).

L115 "sea level variations" should be more appropriate here than "sea level changes"

L152-153, This statement is reasonable, but you used a coarse-resolution (4x4) wind product (WASwind) as your observational reference, could it affect your derivation of "biases" as shown in Fig. 4 and discussions about wind stress curl in the following sections?

L225-229, one important aspect from Fig. 8, not discussed here, is that this sea level trend map (east-west contrast) mainly results from decadal variability rather than represents the long-term trend, as discussed by Bromirski et al. (2011), Zhang and Church (2012), and Hamlington et al. (2014).

L247, by calculating the degrees of freedom in this way, you treat each monthly data point independent from each other, which is not true. You need to calculate "effective" degrees of freedom by considering the autocorrelation of the time series.

Fig. 11, for the left three panels (i.e., a, c, e), it would be ideally to plot the corresponding vector (e.g., JRA55-do bias wind stress vector), rather than the same mean wind stress vector. By doing so, you would also show the meridional wind stress information, which help to understand wind stress curl plots on the right.

L261: you may want to give information of the "five-year mean", over which period?

L288-291, as commented above (L225-229), deriving trends over short periods can be influenced by interannual to decadal variability.

L307, this 0.3 m/s doesn't make sense to me, shouldn't it be around 0.9 m/s (it takes about 6 months for 1st baroclinic Rossby waves to travel across the tropical Pacific basin, which gives a speed of about 0.9 m/s). A simple check of Fig. 16 doesn't support 0.3 m/s.

L376, is it possible to use Johnson et al. (2002) as an observational reference and overplot it in Fig. 19c?

L423, the range for the third stage (Jan-June Year 2) overlaps with the 2nd Stage (Aug Year 1 to Jan Year 2).

L435-437, there are already some studies on the meridional asymmetry, e.g., by McGregor et al. (2012).

L467-468, by designing this regression between DSL and wind stress curl at each grid point, are you implying that sea level responds to mainly local Ekman pumping and wave propagation can be neglected (using the simplified 1st baroclinic Rossby wave model as an example).

References:

Bromirski, P. D., A. J. Miller, R. E. Flick, and G. Auad, 2011: Dynamical suppression of sea level rise along the Pacific Coast of North America: Indications for imminent acceleration, J. Geophys. Res., 116, C07005, doi:10.1029/2010JC006759.

Hamlington, B. D., M.W. Strassburg, R. R. Leben, W. Han and R. S. Nerem and K.-Y. Kim, 2014: Uncovering an anthropogenic sea-level rise signal in the Pacific, Nature Climate Change, 4, 782–785, doi:10.1038/NCLIMATE2307.

McGregor, S., , A. Timmermann, , S. Schneider, , M. F. Stuecker, , and M. H. England, 2012: The effect of the South Pacific convergence zone on the termination of El Niño events and the meridional asymmetry of ENSO. J. Climate, 25, 5566–5586, doi:10.1175/JCLI-D-11-00332.1.

Zhang, X. and J. A. Church, 2012: Sea level trends, interannual and decadal variability in the Pacific Ocean, Geophys. Res. Lett., 39, L21701, doi:10.1029/2012GL053240.

---

## Author Comment (AC1) · 17 Mar 2021

We thank the reviewer's useful comments and suggestions. The questions in the general comments are all listed out separately below. The necessary figures and explanation will be added to the paper to provide robust evidence and makes the paper easier for the reader to follow.

- Cause of the absence of the observed DSL trough around 9N in simulations is made using zonally averaged zonal wind (e.g., Fig.4b). Considering the slanting distribution of the wind convergence zone in the eastern tropical north Pacific, two-dimensional

distribution of wind stress curl anomaly might be more illustrative.

A: A 2D plot of wind stress curl mean state is shown below [figure1] to demonstrate that the main zonal bias seen in figure 4c (manuscript) is located in the north-eastern tropical Pacific. The stronger dipole wind stress curl bias in the north-eastern tropical pacific in CORE might be related to the poorer representation of the slanted feature of ITCZ. However, the DSL mean state bias is not significantly improved in JRA55-do even with the slanted feature (manuscript figure4a, figure3b). This result shows that JRA55-do still has a bias in wind stress which is consistent with the zonal mean analysis in figure 4 (manuscript).

- The seasonal variation of NECC might be more clearly explained by using seasonal evolution of horizontal distribution of DSL in the tropical north Pacific.

A: Agreed. In the original manuscript, we describe that the better seasonal DSL simulation forced by JRA55-do is associates with better timing of NECC. However, we did not show the DSL gradient Hovmöller which can provide a better visual explanation. We will add the following explanation and figure in the updated manuscript. "Due to the better simulation of the Rossby wave propagation which affects the DSL seasonal variation in the narrow zonal bend in JRA55-do, the seasonal variation of meridional DSL gradient is also better represented in the simulation which affects the timing of the NECC. At 180o, the positive anomaly of meridional DSL gradient in CORE during the second half of the year creates an anomalous counter-force which counter-acts the negative meridional DSL gradient that supports NECC strength (figure 2 below). This effect can also be seen in figure19c where NECC in CORE forced simulation decreases in the second half of the year that deviates from the more accurate NECC simulation forced by JRA55-do."

- Add some paragraphs that give quantitative guidance about how the simulated biases may compromise practical assessments of sea level variability based on the OMIP simulations. For example, are the biases of DSL trend comparable to global mean sea

level rise observed in recent years?

A: Giving quantitative guidance on practical sea level variability is a good suggestion. We will add the following basin-scale analysis into the manuscript. "Pacific trend bias over the tropics in CORE is the largest across all three major basins. JRA55-do significantly reduces the bias over the tropical Pacific but has a negative bias in the extratropical region and other basins in both northern and southern hemispheres." The OMIP simulated DSL cannot be used to evaluate the global mean sea level rise due to the definition of DSL does not contain the global mean sea level rising signal, which is stated in line 97-98 of the manuscript.

– Specific comments

- L129: "DSL correlates well with the surface wind stress in the mean state". I think that this expression is somewhat inappropriate. I would suggest something like, "distribution of DSL can be explained well by the surface wind stress in the mean state", if I understand the authors' intention correctly.

A: Correct. We will rephrase as suggested.

- Figures 3 and 4: Given that the locations of wind stress convergence are slanting from southwest to northeast in the eastern north Pacific as shown in Figure 3a, contribution of zonal derivative of the meridional component to wind stress curl might not be neglected and the horizontal distribution of wind stress curl bias would be of interest. Is the bias pattern of either wind stress curl or Sverdrup stream function comparable with the DSL bias shown in Figures 3b and 3c? A bit more detail would clarify the point that should be improved in the forcing dataset.

A: In figure 4c (manuscript), we show both the total wind stress curl (solid line) and contribution of the meridional derivative of the zonal component to the wind stress curl (dashed line). The two lines show little difference for both CORE and JRA55-do (also in the 2D maps shown below). Therefore, we think the zonal derivative of

the meridional component to the wind stress curl is small. Figure4 and 5 below also show the dominating role of zonal wind on wind stress curl and biases. Despite the small contribution, we use the total wind stress curl for all analyses in the study. As for the DSL bias related to the wind stress curl bias, the explanation is included in the response to the first question above and figure 1.

- L.213-215: "We hypothesize that the key reason for the weak NECC is due to both the underestimated zonal wind stress in JRA55-do and a flattening of the DSL trough due to the wind stress curl bias in the northern tropical Pacific found in both CORE and JRA55-do [figure 4b,c]." I think that a bit more detailed explanation is required about how the underestimated zonal wind stress in JRA55-do is related to the weak NECC.

A: The referee is correct. Our original statement of underestimated zonal wind stress in JRA55-do affecting NECC is not explained in detail and cannot be quantified to show the contribution. Since that contribution is not the focus here in the study, we update the sentence by removing the part about zonal wind stress to avoid any confusion.

- L.251: Does the excessive westerly wind trend in CORE and JRA55-do affect simulated features of the global warming hiatus?

A: Since there is no output of surface temperature from OMIP simulation, it is hard to perform a direct comparison between model and observation regarding the hiatus. Based on Kosaka and Xie 2013, sea surface temperature (SST) evolution over the tropical Pacific plays an important role in the global warming hiatus. We compare OISST SST observation with SST in both JRA55-do and CORE forced simulations over the same region used in the study. We find little differences between observation and the two simulations. Therefore, the global warming hiatus should be implicitly included in the simulation.

Kosaka, Yu, and Shang-Ping Xie. "Recent Global-Warming Hiatus Tied to Equatorial Pacific Surface Cooling." Nature 501, no. 7467 (September 2013): 403–7. https://doi.org/10.1038/nature12534.

- L.374-375: How does the improved seasonal variation of DSL in JRA55-do result in the better seasonal variation of NECC?

A: This is explained in detail in the response to the second question above with figure 2.

- L.493-495: "the bias in the wind stress forcing causes biases in the geostrophic current that leads to the flattening of the DSL gradient in the meridional direction". I think that the flattening of the DSL leads to the biases in the geostrophic current. For example, "the bias in the wind stress forcing causes flattening of the DSL gradient in the meridional direction that leads to biases in the geostrophic current".

A: Correct. We will rephrase the sentence as suggested. Technical comments

- L58-59: It would be worth noting here that baroclinic deformation radius in the tropics can be resolved by the 0.25-degree resolution used by GFDL-OM4 as also noted in L.122-123.

A: Agreed. We will add the description about baroclinic deformation radius here, too.

- L180-181: "the heat is not stored in the eastern tropical Pacific but flushed to the western tropical Pacific". I think this is worth noting here that "details are discussed in the next paragraph".

A: Agreed. We will add the sentence to make it easier for readers to follow.

- L.218-220: I think discussion about trend can be moved to somewhere around the paragraphs that discuss DSL trend using Figure 8 in section 4.

A: This arrangement of placing the discussion about trends associated with the mean state of current here is intentional. This is written in a way so the reader can connect the following section (section 4) with the current section (section 3).

- L.252: "extends" should read "extending".

A: Agreed.

- L.266: "(called offsetting factor)". That is actually used as an offsetting factor to correct JRA-55 wind.

A: Agreed. We will change the sentence to avoid the parenthesis.

- L.314: "off" should perhaps read "offset" or "delayed".

A: Agreed. The word will be changed to "delayed".
* * *
[Figure]

**Fig. 1.** (a) The wind stress curl (shading) and wind stress (vector) time mean from WASwind. The wind stress curl bias in (b) the JRA55-do and (c) the CORE.

[Figure]

**Fig. 2.** Hovmöller diagram of monthly climatology showing the meridional mean (2âŮęN to 10 âŮęN) DSL gradient difference by subtracting JRA55-do forced simulation from CORE.

[Figure]

**Fig. 3.** The zonal mean DSL trend bias at all three major basins from JRA55-do (blue) and CORE (orange) forced simulations during 1993-2007.

[Figure]

**Fig. 4.** same as figure 1 with only meridional component.

[Figure]

**Fig. 5.** same as figure 1 with only zonal component

[Figure]

**Fig. 6.** Sea surface temperature mean in the box (20S-20N and west of 180o – eastern boundary) used in the Kosaka and Xie, 2013 study to test the factor affecting the global hiatus.

---

## Author Comment (AC2) · 17 Mar 2021

We thank the reviewer's useful comments and suggestions. To address all reviewer's questions, we listed out all questions and the corresponding answers from us below.

–General Comments

-1. My major comment/concern after reading through this quite lengthy manuscript is that I feel it may be more appropriate for other journals than GMD – a journal focusing on model development. But I have to say it's my personal feeling based on my understanding of GMD versus other journals, and it's really up to the editor to decide. If the

authors agree on this, I suggest the authors can reduce some parts (which don't really show new results), and tighten up the storyline. I guess it would be a better paper.

A: In our opinion, the Geoscience Model Development journal fits well with the purpose of this paper which is focusing on the model evaluation. This is in-line with the "Model evaluation papers" listed in the GMD manuscript types. Our study uses the state-of-the-art ocean model comparing with observations across different time scales which can help us systematically understand the current model biases and further improve the model simulation and development in the future.

-2. Wind stress and wind stress curl are heavily discussed throughout this paper, which is fine. However, the coarse-resolution (4x4 deg) WASwind product is used, and differences between JRA55-do and CORE are defined as "biases". I feel it could be problematic. For example, some features in the wind stress curl map (Fig. 11b) may be "artificial", reflecting the 4x4 grid. I think there should be some better-quality wind products, e.g., based on satellite Scatterometer observations?

A: Yes, we have tried CCMP a multi-mission surface wind analysis that backdates to 1989 and has a higher resolution (0.25x0.25 degree). However, in our study, we can only find significant changes of wind stress trend bias over a longer period (1950-2011) which WASwind provides but not available in CCMP (1989-2011). As an alternative way to verify the WASwind result, we compare the WASwind and CCMP data in the mean, long-term trend, and seasonal variability over the overlapping period. The results are comparable. We did not see significant differences across time scales. Like WASwind, CCMP also shows no statistical significance in long-term trend analysis during the overlapping period. The other important reason we think the coarse resolution does not affect our result is due to the biases in this study are all more than 4 degrees (Figure 4, 4N - 9N, figure 12, figure 16,17 2N-10N). Therefore, we will keep the WASwind analyses in our study. However, we do agree figure 11b,d,f might be somewhat misleading since we are not focusing on all bias that shows on the map that has a smaller spatial scale. The original purpose of the figure is to show the spatial pattern

of the bias that we want to focus on in figure 12. In the revision, we will remove figure 11b,d,f to avoid this confusion. As for the JRA55-do and CORE difference, we do carefully avoid the word "biases" since it is not a comparison with observation. In this part of the analysis, since the observation is limited, we use JRA55-do as a relatively good estimate to evaluate improvement from CORE forced result. As for why JRA55-do can be assumed as a relatively good estimate, it is due to the significant reduction of sea level bias during the period when the observation is available (1993-2007). These reasons are mentioned in our manuscript line 232-237.

–detailed comments

-Line 23 (L23), this sentence is confusing and thus needs to be changed. Sea level variability is not only associated with ocean temperature (heat content), but can also due to halosteric component (ocean salinity) and mass component.

A: We will change to "Sea level in the tropical Pacific is dominated by the ocean heat content variability and long-term trends."

-L31, OMIP-I and II are not defined before. I feel they should be introduced here with a couple of sentences. What are main differences and similarities between them? It may help to state your motivation more clearly. Some material from the 2nd paragraph of Section 2.1 (L63-69) can be moved here.

A: We will move part of Section 2.1 here to strengthen the motivation. "OMIP-I is forced by the CORE dataset of Large and Yeager (2009) and it extends over years 1948-2007 (hereafter, CORE), whereas OMIP-II uses the JRA55-do dataset of Tsujino et al. (2018), which extends over years 1958-2018. " will be moved up.

-L96-97, steric sea level rise is not only due to ocean warming, but also ocean freshening (decreasing salinity).

A: We will change the sentence to "The minus sign on the steric term arises since decreases in density, as from ocean warming or freshening, lead to increases in sea

level."

-L105, change "which relates" to "which mainly relates", since salinity can also play a role, in addition to ocean temperature (or equivalently thermocline depth change).

A: We will change the sentence accordingly.

-L115 "sea level variations" should be more appropriate here than "sea level changes"

A: Agreed. We will change the sentence to "sea level variations".

-L152-153, This statement is reasonable, but you used a coarse-resolution (4x4) wind product (WASwind) as your observational reference, could it affect your derivation of "biases" as shown in Fig. 4 and discussions about wind stress curl in the following sections?

A: To show the coarse resolution is not affecting the analyses we mentioned here, we calculate the coarse resolution WASwind and the fine resolution observation from CCMP (0.25*0.25 degree) over the overlapping period. For the mean-field, it shows a maximum difference between the two products of 40% to WASwind value in the tropical region. This difference can come from the observational technique difference, the resolution difference, or a combined effect. However, the bias we want to show here between 4N to 9N is more than 200% of the observed value in WASwind. This means that even we assume the 40% difference between the WASwind and CCMP is all coming from resolution difference, it is still smaller than the bias we want to show here.

-L225-229, one important aspect from Fig. 8, not discussed here, is that this sea level trend map (east-west contrast) mainly results from decadal variability rather than represents the long-term trend, as discussed by Bromirski et al. (2011), Zhang and Church (2012), and Hamlington et al. (2014).

A: Agreed. We will add in the decadal variability studies since this would also affect the trend analysis over a short time period.

-L247, by calculating the degrees of freedom in this way, you treat each monthly data point independent from each other, which is not true. You need to calculate "effective" degrees of freedom by considering the autocorrelation of the time series.

A: Agreed. We will update the description here related to the degree of freedom. The results from the analyses have little to no change after updating the degree of freedom.

-Fig. 11, for the left three panels (i.e., a, c, e), it would be ideally to plot the corresponding vector (e.g., JRA55-do bias wind stress vector), rather than the same mean wind stress vector. By doing so, you would also show the meridional wind stress information, which help to understand wind stress curl plots on the right.

A: Yes, we will change panel c,e to vector showing the bias.

-L261: you may want to give information of the "five-year mean", over which period?

A: It is 2000-2004 [Large and Yeager, 2009]. We will add this info in the revision.

-L288-291, as commented above (L225-229), deriving trends over short periods can be influenced by interannual to decadal variability.

A: Yes, we agree. We will add the citation and mentioned the possible contribution from decadal variability based on past studies.

-L307, this 0.3 m/s doesn't make sense to me, shouldn't it be around 0.9 m/s (it takes about 6 months for 1st baroclinic Rossby waves to travel across the tropical Pacific basin, which gives a speed of about 0.9 m/s). A simple check of Fig. 16 doesn't support 0.3 m/s.

A: The calculation is done by looking at the contour (dashed) in figure 16a,c,e (or figure 18f which shows the 20C isotherm depth) starting from 110W to 150E (total 110 degree longitude) from January to December. The number we used to calculate the speed is as follows. 110 [degree longitude] * 100000 [m/degree] / (12*30*24*60*60) [s] $\sim$ 0.35 m/s This baroclinic Rossby wave and its non-dispersive phase speed are also mentioned

in Meyers which matches with our number [1979]. We will add the above information in the revised manuscript for clarification. Meyers, G. (1979). On the Annual Rossby Wave in the Tropical North Pacific Ocean. Journal of Physical Oceanography, 9(4), 663–674. https://doi.org/10.1175/1520-0485(1979)009<0663:OTARWI>2.0.CO;2

-L376, is it possible to use Johnson et al. (2002) as an observational reference and overplot it in Fig. 19c?

A: Johnson et al. [2002] only mentioned the maximum (peak) month in their study. We could not find a whole seasonal cycle (monthly values) available.

-L423, the range for the third stage (Jan-June Year 2) overlaps with the 2nd Stage (Aug Year 1 to Jan Year 2).

A: Right, this is an intentional overlap since Jan Year2 is when the maximum ONI happens. The main purpose of this analysis here is to show that the difference of 6 months mean before and after the maximum ONI is very different. We also tried to shift one month for both stage3 and stage4. It shows little change to result.

-L435-437, there are already some studies on the meridional asymmetry, e.g., by McGregor et al. (2012).

A: Thank you for letting us know about this study. We will cite this study in this part of the analysis.

-L467-468, by designing this regression between DSL and wind stress curl at each grid point, are you implying that sea level responds to mainly local Ekman pumping and wave propagation can be neglected (using the simplified 1st baroclinic Rossby wave model as an example).

A: No. The regression map is trying to emphasize the strong correlation in the region of interest (black dashed box in figure 22 23 24 25) mentioned in the analyses when comparing with other regions. It does not imply the wave propagation during ENSO can be neglected. Wave propagation still plays a central role in the ENSO dynamics.

---

## Author Response (AR1)

**We thank the reviewer's useful comments and suggestions. The questions in the general comments are all listed out separately below. The necessary figures and explanation will be added to the paper to provide robust evidence and makes the paper easier for the reader to follow.**

Cause of the absence of the observed DSL trough around 9N in simulations is made using zonally averaged zonal wind (e.g., Fig.4b). Considering the slanting distribution of the wind convergence zone in the eastern tropical north Pacific, two-dimensional distribution of wind stress curl anomaly might be more illustrative.

A: A 2D plot of wind stress curl mean state is shown below [figure1] to demonstrate that the main zonal bias seen in figure 4c (manuscript) is located in the north-eastern tropical Pacific. The stronger dipole wind stress curl bias in the north-eastern tropical pacific in CORE might be related to the poorer representation of the slanted feature of ITCZ. However, the DSL mean state bias is not significantly improved in JRA55-do even with the slanted feature (manuscript figure4a, figure3b). This result shows that JRA55-do still has a bias in wind stress which is consistent with the zonal mean analysis in figure 4 (manuscript).

Figure 1. (a) The shaded color shows the wind stress curl time mean (1993-2007) from observation (WASwind) and the vector field shows the wind stress time mean (see the  $0.1 \text{ N m}^{-2}$  vector in the upper left for scale). The shaded color shows wind stress curl bias in (b) the JRA55-do and (c) the CORE during the 1993-2007 period.

The seasonal variation of NECC might be more clearly explained by using seasonal evolution of horizontal distribution of DSL in the tropical north Pacific.

A: Agreed. In the original manuscript, we describe that the better seasonal DSL simulation forced by JRA55-do is associates with better timing of NECC. However, we did not show the DSL gradient Hovmöller which can provide a better visual explanation. We will add the following explanation and figure in the updated manuscript. "Due to the better simulation of the Rossby wave propagation which affects the DSL seasonal variation in the narrow zonal bend in JRA55-do, the seasonal variation of meridional DSL gradient is also better represented in the simulation which affects the timing of the NECC. At 180°, the positive anomaly of meridional DSL gradient in CORE during the second half of the year creates a anomalous counter-force which counter-acts the negative meridional DSL gradient that supports NECC strength (figure 2 below). This effect can also be seen in figure 19c where NECC in CORE forced simulation decreases in the second half of the year that deviates from the more accurate NECC simulation forced by JRA55-do."